# Cryo-EM structures and functional characterization of the murine lipid scramblase TMEM16F

Carolina Alvadia[1†], Novandy K Lim[1†], Vanessa Clerico Mosina[2†], Gert T Oostergetel[2], Raimund Dutzler[1*], Cristina Paulino[2*]

[1]Department of Biochemistry, University of Zurich, Zurich, Switzerland; [2]Department of Structural Biology at the Groningen Biomolecular Sciences and Biotechnology Institute, University of Groningen, Groningen, Netherlands

**Abstract** The lipid scramblase TMEM16F initiates blood coagulation by catalyzing the exposure of phosphatidylserine in platelets. The protein is part of a family of membrane proteins, which encompasses calcium-activated channels for ions and lipids. Here, we reveal features of murine TMEM16F (mTMEM16F) that underlie its function as a lipid scramblase and an ion channel. The cryo-EM data of mTMEM16F in absence and presence of $Ca^{2+}$ define the ligand-free closed conformation of the protein and the structure of a $Ca^{2+}$-bound intermediate. Both conformations resemble their counterparts of the scrambling-incompetent anion channel mTMEM16A, yet with distinct differences in the region of ion and lipid permeation. In conjunction with functional data, we demonstrate the relationship between ion conduction and lipid scrambling. Although activated by a common mechanism, both functions appear to be mediated by alternate protein conformations that are at equilibrium in the ligand-bound state.
DOI: https://doi.org/10.7554/eLife.44365.001

*For correspondence:
dutzler@bioc.uzh.ch (RD);
c.paulino@rug.nl (CP)

†These authors contributed equally to this work

Competing interests: The authors declare that no competing interests exist.

## Introduction

Lipid scramblases facilitate the movement of lipids between both leaflets of the bilayer (*Bevers and Williamson, 2016*; *Nagata et al., 2016*; *Williamson, 2015*), thereby dissipating the asymmetry present in the membrane. In contrast to ATP-dependent lipid flippases and floppases, scramblases are generally non-selective and do not require the input of energy (*Pomorski and Menon, 2016*). Lipid scrambling changes the properties of the bilayer and thus contributes to membrane synthesis, fusion and repair and the release of extracellular vesicles (*Whitlock and Hartzell, 2017*). The resulting exposure of negatively charged phospholipids to the outside is sensed by receptors that in turn initiate cellular responses such as blood coagulation and apoptosis (*Nagata et al., 2016*). In platelets, scrambling is mediated by the protein TMEM16F, which catalyzes the externalization of phosphatidylserine (PS) in response to an increase in the intracellular $Ca^{2+}$ concentration to trigger blood clotting (*Bevers and Williamson, 2016*; *Suzuki et al., 2013*). Splicing mutations that lead to a truncated version of TMEM16F were found to cause Scott syndrome, a severe bleeding disorder in humans and dogs (*Brooks et al., 2015*; *Castoldi et al., 2011*; *Lhermusier et al., 2011*). TMEM16F belongs to the TMEM16 protein family, whose members either function as $Ca^{2+}$ activated anion-selective channels (*Caputo et al., 2008*; *Schroeder et al., 2008*; *Yang et al., 2008*) or as $Ca^{2+}$ activated lipid scramblases (*Brunner et al., 2016*; *Falzone et al., 2018*; *Whitlock and Hartzell, 2017*). The structure of the fungal TMEM16 homologue from *Nectria haematococca* (nhTMEM16), determined by X-ray crystallography, has defined the general architecture of the family and provided insight into the mechanism of lipid translocation (*Brunner et al., 2014*). In nhTMEM16, each subunit of the homodimeric protein contains a membrane-accessible polar furrow termed the 'subunit cavity',

which provides a suitable pathway for the polar lipid headgroups on their way across the hydrophobic core of the bilayer (*Bethel and Grabe, 2016*; *Brunner et al., 2014*; *Jiang et al., 2017*; *Lee et al., 2018*; *Stansfeld et al., 2015*). This process closely resembles the 'credit card mechanism' for scrambling, which was previously postulated based on theoretical considerations (*Pomorski and Menon, 2006*). Conversely, single particle cryo-electron microscopy (cryo-EM) structures of murine TMEM16A (mTMEM16A), which instead of transporting lipids solely facilitates selective anion permeation (*Dang et al., 2017*; *Paulino et al., 2017a*; *Paulino et al., 2017b*), revealed the structural differences that underlie the distinct function of this branch of the TMEM16 family. In mTMEM16A, the rearrangement of an α-helix that lines one edge of the 'subunit cavity' in nhTMEM16 seals the membrane-accessible furrow, resulting in the formation of a protein-enclosed aqueous pore that is for a large part shielded from the bilayer. In both proteins, binding of $Ca^{2+}$ mediates the activation of the permeation region contained in each subunit, which in mTMEM16A was shown to act as an independent entity (*Jeng et al., 2016*; *Lim et al., 2016*). Whereas the structures of nhTMEM16 and mTMEM16A have defined the architecture of two distantly related members of the family, TMEM16F appears, with respect to phylogenetic relationships, as intermediate between the two proteins. Although working as lipid scramblase (*Suzuki et al., 2010*; *Watanabe et al., 2018*), it is closer related to the ion channel mTMEM16A than to nhTMEM16 (*Brunner et al., 2016*; *Falzone et al., 2018*; *Whitlock and Hartzell, 2017*). Moreover, whereas scrambling-related ion conduction was found to be a feature of several family members (*Lee et al., 2016*; *Malvezzi et al., 2018*; *Whitlock and Hartzell, 2016*), TMEM16F is the only lipid scramblase for which rapidly activated calcium-dependent currents were recorded in excised patches (*Yang et al., 2012*). To better understand how the small sequence differences in TMEM16F give rise to its distinct functional properties, we determined the structure of murine TMEM16F (mTMEM16F) by cryo-EM in $Ca^{2+}$-bound and $Ca^{2+}$-free states, both in a detergent and in a lipid environment. In parallel, we characterized the lipid transport properties of mTMEM16F *in vitro* after reconstitution of the protein into liposomes, as well as ion conduction properties in transfected cells by electrophysiology. Collectively, our study reveals the architecture of mTMEM16F, defines conformational changes upon ligand binding and suggests potential mechanisms for ion and lipid conduction. In the most plausible scenario, both transport processes are activated by the same mechanism and mediated by distinct protein conformations, which are at equilibrium in a calcium-bound state.

## Results

### Functional properties of mTMEM16F

In our study, we explored the relationship between the structure of mTMEM16F, which shares a sequence identity of over 90% with its human orthologue, to its diverse functional properties. For this purpose, we have expressed mTMEM16F in HEK293 cells, purified it in the detergent digitonin in the absence of $Ca^{2+}$ and, when indicated, added $Ca^{2+}$ briefly before experiments (*Figure 1—figure supplement 1A,B*). To confirm that the purified protein has retained its function as a $Ca^{2+}$-activated lipid scramblase, we investigated lipid transport with proteoliposomes using an assay that was previously established for fungal TMEM16 scramblases (*Malvezzi et al., 2013*; *Ploier and Menon, 2016*). This assay monitors the irreversible reduction of fluorescent lipids upon addition of the membrane-impermeable reagent dithionite to the outside of liposomes. The fluorescence rapidly decays to about half of its initial value in the absence of scrambling due to the bleaching of lipids already facing the outside, and it decreases further in a time-dependent manner in cases where a scramblase catalyzes lipid transport from the inner to the outer leaflet (*Figure 1—figure supplement 1C*). Our data reveal a $Ca^{2+}$-induced activity, which, although dependent on the composition of proteoliposomes, facilitates transport of lipids with different headgroups, consistent with the broad lipid selectivity that was described for mTMEM16F and other TMEM16 scramblases (*Brunner et al., 2014*; *Malvezzi et al., 2013*; *Suzuki et al., 2013*) (*Figure 1—figure supplement 1C–E*). Activation of scrambling is fast, as evidenced by the immediate decay of the fluorescence upon addition of $Ca^{2+}$ to the outside of mTMEM16F containing proteoliposomes (*Figure 1—figure supplement 1F*), and saturates with an $EC_{50}$ for $Ca^{2+}$ of about 1 μM (*Figure 1A,B*). Thus, our experiments demonstrate the function of purified and reconstituted mTMEM16F as a $Ca^{2+}$-activated lipid scramblase.

Besides its capability to transport lipids, mTMEM16F was also reported to act as an ion channel (*Kunzelmann et al., 2014*; *Scudieri et al., 2015*; *Yang et al., 2012*; *Yu et al., 2015*). To recapitulate these properties, we have studied ion conduction in inside-out patches excised from HEK293 cells expressing mTMEM16F. As described previously, mTMEM16F mediates currents that are activated in response to elevated intracellular $Ca^{2+}$ concentrations (*Yang et al., 2012*) (*Figure 1C,D*; *Figure 1—figure supplement 1G*). Analogous to the $Ca^{2+}$-dependence of scrambling, the activation of ion conduction is rapid and currents (recorded at 80 mV) saturate with an $EC_{50}$ of 4–7 µM. Similar results were obtained from the stable cell-line expressing mTMEM16F, used for reconstitution and structure determination, and from cells that were transfected with a construct containing a C-terminal fusion to YFP, which was used for the further characterization of mTMEM16F by electrophysiology (*Figure 1D*; *Figure 1—figure supplement 1H*). The Hill coefficient of about 1.7 for both mTMEM16F constructs emphasizes the cooperativity of the process. Unlike for the anion-selective mTMEM16A, the currents are slightly selective for cations over anions and they retain their strong outward rectification in the entire ligand-concentration range (*Figure 1C*; *Figure 1—figure supplement 1G,I*). Whereas the mild rectification of instantaneous currents is likely a consequence of conduction through an asymmetric pore (*Paulino et al., 2017b*), the presence of time-dependent relaxations in response to changes in the transmembrane voltage and the increased rectification at steady-state reflect a voltage-dependent transition that is not obviously linked to ligand binding (*Figure 1C*; *Figure 1—figure supplement 1G*). As the activation of ion conduction and lipid scrambling occurs in a very similar ligand concentration range, both processes likely rely on the same mechanism of $Ca^{2+}$-activation.

## Structural characterization

Since mTMEM16F combines the functional properties of a lipid scramblase and an ion channel, we were interested in the architectural features that distinguishes it from the anion-selective channel mTMEM16A and the fungal scramblase nhTMEM16. For that purpose, we determined structures of mTMEM16F in the absence and presence of $Ca^{2+}$ (*Table 1*). Structures of the protein in the detergent digitonin were determined for $Ca^{2+}$-bound and $Ca^{2+}$-free states by cryo-EM at 3.2 Å and 3.6 Å, respectively (*Figure 2*; *Figure 2—figure supplements 1* and *2*; *Videos 1* and *2*). Both structures are

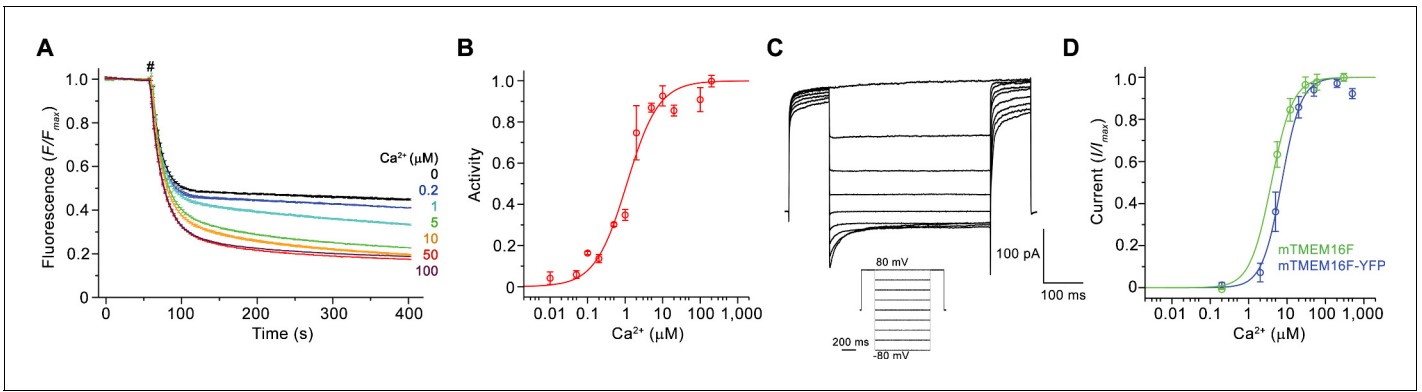

**Figure 1.** Functional characterization of mTMEM16F. (**A**) $Ca^{2+}$-dependence of scrambling activity in mTMEM16F-containing proteoliposomes. Traces depict fluorescence decrease of tail-labeled NBD-PE lipids after addition of dithionite (#) at different $Ca^{2+}$ concentrations. Data show averages of three technical replicates. (**B**) $Ca^{2+}$ concentration-response relationship of mTMEM16F scrambling ($EC_{50}$ = 1.1 µM, n = 1.0). Data show mean of six independent experiments from three protein reconstitutions. (**C**) Representative current traces at 200 µM $Ca^{2+}$ recorded from inside-out patches excised from HEK293 cells transiently expressing a mTMEM16F-YFP fusion construct. Inset shows voltage protocol. (**D**) $Ca^{2+}$ concentration-response relationship of currents recorded from inside-out patches of a stable HEK293 cell-line expressing mTMEM16F (green, $EC_{50}$ = 3.9 µM, n = 1.6) and from HEK293 cells transiently expressing a mTMEM16F-YFP fusion construct (blue, $EC_{50}$ = 7.2 µM, n = 1.7). Data show mean of five (stable cell-line) and seven biological replicates (transfected cells). B, D Solid lines show fit to a Hill equation. A, B, D, errors are s.e.m..

DOI: https://doi.org/10.7554/eLife.44365.002

The following figure supplement is available for figure 1:

**Figure supplement 1.** Biochemistry and functional characterization.
DOI: https://doi.org/10.7554/eLife.44365.003

**Table 1.** Cryo-EM data collection, refinement and validation statistics.

| | mTMEM16F dig, +Ca²⁺ (EMDB 4611, PDBID 6QP6) | mTMEM16F dig, -Ca²⁺ (EMDB 4612, PDBID 6QPB) | mTMEM16F 2N2, +Ca²⁺ (EMDB 4613, PDBID 6QPC) | mTMEM16F 2N2, -Ca²⁺ (EMDB 4614, PDBID 6QPI) |
|---|---|---|---|---|
| **Data collection and processing** | | | | |
| Microscope | FEI Talos Arctica | FEI Talos Arctica | FEI Talos Arctica | FEI Talos Arctica |
| Camera | Gatan K2 Summit + GIF | Gatan K2 Summit + GIF | Gatan K2 Summit + GIF | Gatan K2 Summit + GIF |
| Magnification | 49,407 | 49,407 | 49,407 | 49,407 |
| Voltage (kV) | 200 | 200 | 200 | 200 |
| Exposure time frame/total (s) | 0.15/9 | 0.15/9 | 0.15/9 | 0.15/9 |
| Number of frames per image | 60 | 60 | 60 | 60 |
| Electron exposure (e–/Å²) | 52 | 52 | 52 | 52 |
| Defocus range (μm) | −0.5 to −2.0 | −0.5 to −2.0 | −0.5 to −2.0 | −0.5 to −2.0 |
| Pixel size (Å) | 1.012 | 1.012 | 1.012 | 1.012 |
| Box size (pixels) | 220 | 220 | 256 | 220 |
| Symmetry imposed | C2 | C2 | C2 | C2 |
| Initial particle images (no.) | 1,348,247 | 1,314,676 | 1,019,012 | 1,593,115 |
| Final particle images (no.) | 219,302 | 194,284 | 186,487 | 280,891 |
| Map resolution (Å) 0.143 FSC threshold | 3.18 | 3.64 | 3.54 | 3.27 |
| Map resolution range (Å) | 2.9–4.5 | 3.2–5 | 3.3–5 | 3.0–4.2 |
| **Refinement** | | | | |
| Initial model used | PDBID 5OYB | PDBID 6QP6 | PDBID 6QP6 | PDBID 6QPB |
| Model resolution (Å) FSC threshold | 3.3 | 3.8 | 4.0 | 7.0 |
| Model resolution range (Å) | 80–3.2 | 80–3.6 | 80–3.5 | 80–3.3 |
| Map sharpening *B* factor (Å²) | −94 | −139 | −121 | −131 |
| Model composition | | | | |
| Nonhydrogen atoms | 12144 | 12136 | 9060 | 2288 |
| Protein residues | 1462 | 1460 | 1138 | 462 |
| Ligands | 8 | 2 | 6 | - |
| *B* factors (Å²) | | | | |
| Protein | 53.3 | 52 | 72.7 | 7.8 |
| Ligand | 39.1 | 34.8 | 54.6 | - |
| R.m.s. deviations | | | | |
| Bond lengths (Å) | 0.009 | 0.007 | 0.005 | 0.004 |
| Bond angles (°) | 1 | 0.9 | 0.87 | 0.655 |
| Validation | | | | |
| MolProbity score | 1.36 | 1.23 | 1.3 | 1.17 |
| Clashscore | 6.5 | 4.5 | 5.6 | 2.4 |
| Poor rotamers (%) | 0.46 | 0.3 | 0.22 | 0 |
| Ramachandran plot | | | | |
| Favored (%) | 98.6 | 98.4 | 98.4 | 97.2 |
| Allowed (%) | 1.4 | 1.6 | 1.6 | 2.8 |
| Disallowed (%) | 0 | 0 | 0 | 0 |

DOI: https://doi.org/10.7554/eLife.44365.012

very similar, except for conformational differences at the Ca²⁺-binding site and the 'subunit cavity' (**Figure 2C**).

To investigate the effect of a lipid bilayer on the structural properties of mTMEM16F, we reconstituted the protein into large 2N2 lipid nanodiscs with a lipid composition that retains scrambling activity, although with slower kinetics (**Figure 1—figure supplement 1D**), and collected cryo-EM data of both states in a membrane-like environment (**Figure 2—figure supplements 3** and **4**). While of similar nominal resolution to the data obtained in detergent, the preferred orientation of the

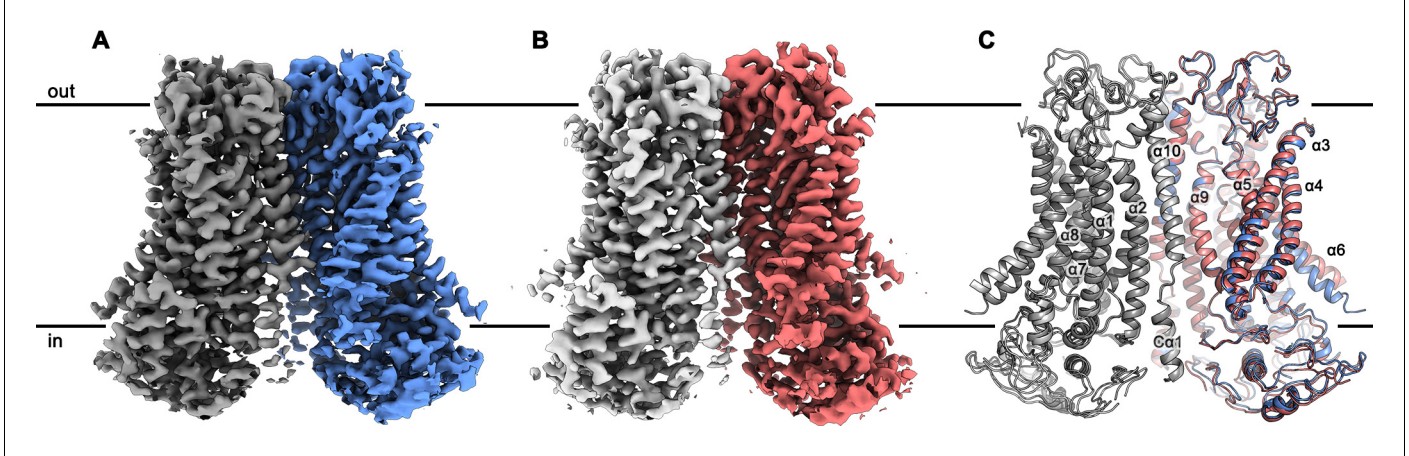

**Figure 2.** mTMEM16F structures. Cryo-EM map of mTMEM16F in digitonin in presence (**A**) and absence (**B**) of calcium at 3.2 Å and 3.6 Å, respectively. (**C**) Cartoon representation of a superposition of the $Ca^{2+}$-bound (blue and grey) and $Ca^{2+}$-free mTMEM16F (magenta and light grey) dimer. Both structures superimpose with a root mean square deviation (RMSD) of 1.2 Å. Transmembrane helices are labelled and the membrane boundary is indicated.

DOI: https://doi.org/10.7554/eLife.44365.004

The following figure supplements are available for figure 2:

**Figure supplement 1.** Structure determination of mTMEM16F in complex with $Ca^{2+}$ in digitonin.
DOI: https://doi.org/10.7554/eLife.44365.005

**Figure supplement 2.** Structure determination of mTMEM16F in absence of $Ca^{2+}$ in digitonin.
DOI: https://doi.org/10.7554/eLife.44365.006

**Figure supplement 3.** Structure determination of mTMEM16F in complex with $Ca^{2+}$ in nanodisc.
DOI: https://doi.org/10.7554/eLife.44365.007

**Figure supplement 4.** Structure determination of mTMEM16F in absence of $Ca^{2+}$ in nanodiscs.
DOI: https://doi.org/10.7554/eLife.44365.008

**Figure supplement 5.** Cryo-EM Density Maps.
DOI: https://doi.org/10.7554/eLife.44365.009

**Figure supplement 6.** Superpositions of the four final cryo-EM maps.
DOI: https://doi.org/10.7554/eLife.44365.010

**Figure supplement 7.** Cryo-EM density of lipids at the dimer interface.
DOI: https://doi.org/10.7554/eLife.44365.011

protein in nanodiscs on the grids resulted in anisotropic cryo-EM maps, in which some of the details are blurred (*Figure 2—figure supplements 3–5*). This problem is less pronounced in the data obtained in nanodiscs in presence of $Ca^{2+}$ than in the dataset of the $Ca^{2+}$-free sample. Nonetheless, both datasets are of sufficient quality to allow for a comparison of corresponding states of mTMEM16F in a detergent and lipid environment. For most parts, these structures are virtually indistinguishable except for α3 and α4, which show a larger tilt towards α6 in the cryo-EM maps of nanodisc samples, a feature that is more prominent in the $Ca^{2+}$-free state (*Figure 2—figure supplement 6*). Thus, our data reveal a general equivalence of structures in detergent and in a membrane environment but they also hint at context-dependent differences in a region that constitutes the ion and lipid permeation path (*Figure 2—figure supplements 5* and *6*). In addition, we were interested whether we could identify any structural flexibility within each of the four datasets. While no heterogeneity was found in the data in absence of $Ca^{2+}$ both in detergent and nanodiscs (*Figure 2—figure supplements 2I* and *4I*), we were able to identify classes with slightly altered conformations for α3 and α4 in the detergent and nanodiscs samples in presence of $Ca^{2+}$ (*Figure 2—figure supplements 1I* and *3I*). These differences are more pronounced in the $Ca^{2+}$-bound nanodisc sample and could point towards a further structural transition of these α-helices from their current position into an open conformation of the 'subunit cavity', as observed for the fungal scramblases nhTMEM16 (*Kalienkova et al., 2019*) and afTMEM16 (*Falzone et al., 2019*) and for the human TMEM16K (hTMEM16K, *Bushell et al., 2018*). However, despite the conformational spread of α3 and α4, no

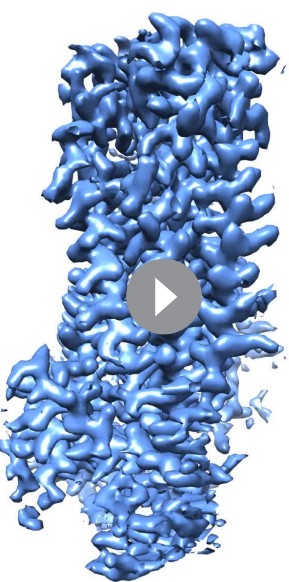

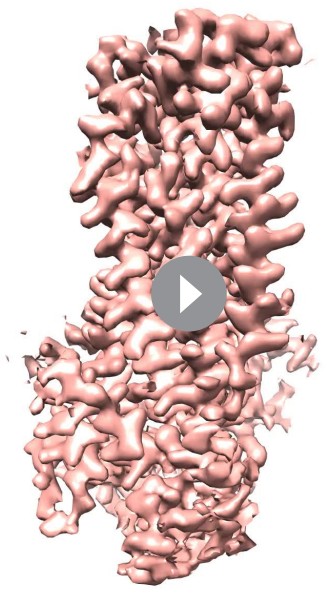

**Video 1.** Structure of mTMEM16F in complex with Ca$^{2+}$. Shown is the cryo-EM density map of mTMEM16F obtained in complex with Ca$^{2+}$ in digitonin superimposed on the refined structure. For clarity, only a single subunit is displayed. The cryo-EM map is depicted as blue surface, the model as blue sticks, and calcium ions as green spheres.
DOI: https://doi.org/10.7554/eLife.44365.013

**Video 2.** Structure of mTMEM16F in absence of Ca$^{2+}$. Shown is the cryo-EM density map of the mTMEM16F obtained in a Ca$^{2+}$-free state in digitonin superimposed on the refined structure. For clarity, only a single subunit is displayed. The cryo-EM map is depicted as magenta surface and the model as red sticks.
DOI: https://doi.org/10.7554/eLife.44365.014

class showing an open conformation of the 'subunit cavity' was identified in our data (*Figure 2—figure supplement 6*).

Unassigned densities near the dimer interface present in all cryo-EM maps most likely account for tightly bound lipids (*Figure 2—figure supplement 7*). Although the potential functional relevance of these lipids is still unclear, similar densities were found in other mammalian TMEM16 proteins, such as mTMEM16A (*Paulino et al., 2017a*; *Dang et al., 2017*) and hTMEM16K (*Bushell et al., 2018*), but not in fungal homologs. In contrast to mTMEM16A (*Paulino et al., 2017b*), nhTMEM16 (*Kalienkova et al., 2019*) and afTMEM16 (*Falzone et al., 2019*) no obvious distortion of the detergent micelle or the lipid nanodiscs was observed in the maps of mTMEM16F (*Figure 2—figure supplements 1H*, *2H*, *3H* and *4H*). Yet, in light of the structural similarity between family members, a destabilizing effect of the protein in its active conformation on the membrane is conceivable. Due to the superior quality of the structures of mTMEM16F in detergent, they are used for further structural analysis.

## Relationship to other TMEM16 structures

With a pairwise sequence identity of 38%, mTMEM16F is closer to the anion channel mTMEM16A (*Figure 3—figure supplement 1*) than to the more distantly related fungal scramblase nhTMEM16 and its mammalian orthologue hTMEM16K, with which it shares sequence identities of 21% and 25%, respectively. It is thus not surprising that, with respect to its general architecture, mTMEM16F closely resembles mTMEM16A (*Figure 3—figure supplement 2*). The similarity to mTMEM16A extends to all parts of the protein, including the cytoplasmic and extracellular domains, and it is particularly pronounced in the Ca$^{2+}$-bound conformations, which superimpose with a root mean square deviation (RMSD) of 2.1 Å (*Figure 3—figure supplement 2A*). The RMSD decreases to 1 Å when relating Cα-positions of the transmembrane domain of a single subunit (encompassing 298 Cα-positions, which are also used in superpositions with nhTMEM16, *Figure 3A,B*). The potential correspondence of both structures of mTMEM16F to functional states of a scramblase is best illustrated in

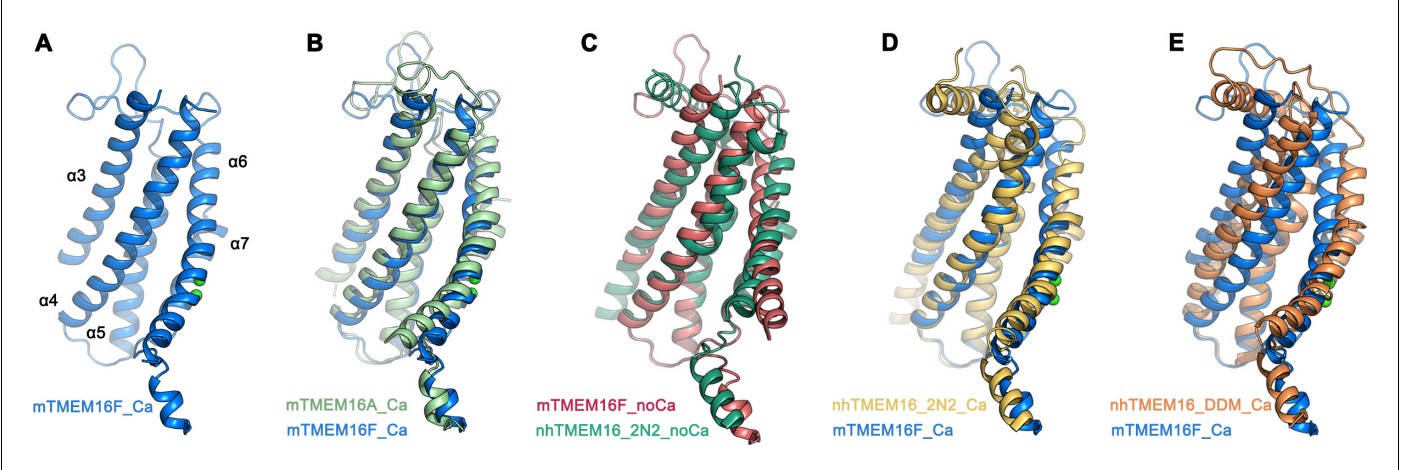

**Figure 3.** Comparison of the 'subunit cavity' of TMEM16 homologues. (A) 'Subunit cavity' in the Ca²⁺-bound mTMEM16F structure composed of α-helices 3–7 (blue). (B) Superposition of the 'subunit cavity' between the Ca²⁺-bound structures of mTMEM16F (blue) and the anion channel mTMEM16A (PDBID 5OYB (*Paulino et al., 2017a*), green). (C) Superposition of the 'subunit cavity' between the Ca²⁺-free structure of mTMEM16F (magenta) and the Ca²⁺-free structure of the fungal scramblase nhTMEM16 in nanodiscs (PDBID 6QM4 (*Kalienkova et al., 2019*), teal) displaying a closed state of the scramblase. (D) Superposition of the Ca²⁺-bound structure of mTMEM16F (blue) with a Ca²⁺-bound conformation of nhTMEM16 in nanodiscs (PDBID 6QMA (*Kalienkova et al., 2019*), yellow) displaying an intermediate state of the scramblase. (E) Superposition of the Ca²⁺-bound structure of mTMEM16F (blue) with a Ca²⁺-bound conformation of nhTMEM16 in detergents (PDBID 6QM5 (*Kalienkova et al., 2019*), orange) displaying an open state of the scramblase.

DOI: https://doi.org/10.7554/eLife.44365.015

The following figure supplements are available for figure 3:

**Figure supplement 1.** Sequence alignment.

DOI: https://doi.org/10.7554/eLife.44365.016

**Figure supplement 2.** Superposition of TMEM16 dimers.

DOI: https://doi.org/10.7554/eLife.44365.017

their comparison to nhTMEM16, for which closed, intermediate and open conformations have been determined (*Kalienkova et al., 2019*). The equivalence of the Ca²⁺-free conformation of mTMEM16F with an inactive ligand-free state of a TMEM16 scramblase is evident in the low RMSD of 2.0 Å of a superposition with the Ca²⁺-free conformation of nhTMEM16 in nanodiscs (*Figure 3C*). In the case of Ca²⁺-bound structures, the relationship of mTMEM16F to the intermediate conformation of nhTMEM16 observed in nanodiscs is, with an RMSD of 1.9 Å, much closer than to the open conformation of the same protein, where the RMSD of 3.1 Å underlines their nonequivalence (*Figure 3D,E*; *Figure 3—figure supplement 2B*). The large differences originate from the diversity of conformations of the subunit cavity observed for Ca²⁺-bound states of TMEM16 proteins. The cavity is accessible to the membrane in the case of the Ca²⁺-bound open state of nhTMEM16 (*Brunner et al., 2014*; *Kalienkova et al., 2019*) and equivalent conformations of its close homologues afTMEM16 (*Falzone et al., 2019*) and hTMEM16K (*Bushell et al., 2018*), but not in the Ca²⁺-bound intermediate and closed conformations of nhTMEM16 (*Kalienkova et al., 2019*) and the observed Ca²⁺-bound conformation of mTMEM16F (*Figure 3C–E*). It is thus uncertain whether the here captured Ca²⁺-bound structure of mTMEM16F shows a partly activated state, which is incapable of scrambling lipids, akin to the intermediate state of nhTMEM16, or whether the scrambling mechanism of mTMEM16F differs from nhTMEM16 and its mammalian orthologues. In light of the ion permeation properties of mTMEM16F and its similarity to mTMEM16A, it is conceivable that the here observed Ca²⁺-bound structure is close to a conducting conformation for ions. In that respect, the question is pertinent why mTMEM16F but not mTMEM16A catalyzes lipid movement.

## Architecture of the 'subunit cavity'

The structure of the putative catalytic unit of mTMEM16F in a Ca²⁺-bound conformation shows a strong resemblance to the ion permeation path in mTMEM16A, where an hourglass-shaped pore contains two aqueous vestibules on either side of the membrane that are bridged by a narrow neck

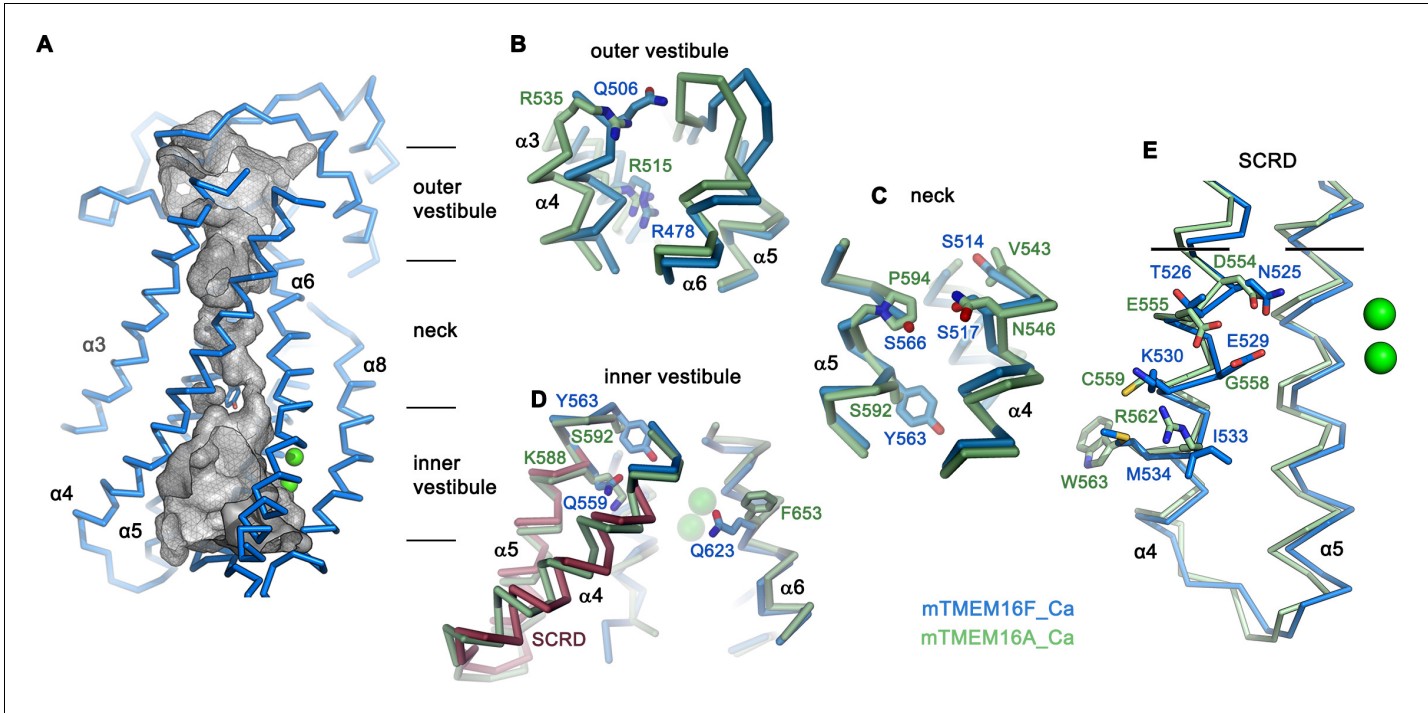

**Figure 4.** Pore region of mTMEM16F. (**A**) 'Subunit cavity' in the Ca$^{2+}$-bound state of mTMEM16F showing the putative ion conduction pore. Grey mesh shows pore surface (sampled with probe radius of 1 Å). (**B** to **D**) Superposition of selected pore regions of mTMEM16F (blue) and the anion channel mTMEM16A (green, remodeled version of PDBID 5OYB, see Materials and methods) along the extracellular vestibule (**B**), the neck (**C**) and the intracellular vestibule, with the 35 amino acid long scrambling domain (SCRD) on α4 and α5 of mTMEM16F highlighted in brown (**D**). (**E**) View of the SCRD rotated by about 90° relative to D. The SCRD boundary is indicated by black lines. A-E, Selected residues are displayed as sticks, bound calcium ions as green spheres.

DOI: https://doi.org/10.7554/eLife.44365.018

The following figure supplement is available for figure 4:

**Figure supplement 1.** Structural features of the pore region and electrostatics.

DOI: https://doi.org/10.7554/eLife.44365.019

(*Paulino et al., 2017a*) (*Figure 4—figure supplement 1A*). A related architecture is observed in mTMEM16F, where closer interactions between α-helices and the presence of bulky residues along the path further constrict the pore below the size of permeating ions (*Figure 4A–E*; *Figure 4—figure supplement 1A,B*). Thus, it is unclear whether the observed conformation of mTMEM16F could facilitate ion conduction. In mTMEM16A, a strong positive electrostatic potential throughout the pore, originating from an excess of cationic residues, determines its selectivity for anions (*Paulino et al., 2017a*). Due to an altered distribution of charges in the pore region of mTMEM16F, the potential is less positive on both entrances to the narrow neck, which is consistent with the poor selectivity of currents passed by this protein (*Figure 4—figure supplement 1C*; *Figure 1—figure supplement 1I*).

Whereas in the extracellular vestibule the uncharged Gln 506, which has replaced an arginine at the equivalent position in mTMEM16A, lowers the positive charge density in the outer mouth of the pore, Arg 478 at the extracellular boundary to the neck is conserved (*Figure 4B*). The neck of mTMEM16F exhibits a similar amphiphilic character as found in mTMEM16A but it contains several changes, which decrease its hydrophobicity (*Figure 4C*). Further towards the cytoplasm, the bulky Tyr 563 lowers the pore diameter of mTMEM16F to 1.6 Å (*Figure 4 C,D*; *Figure 4—figure supplement 1B*). Finally, at the boundary to the wide intracellular vestibule, Gln 559 replaces a lysine at the equivalent position in mTMEM16A, thereby removing another positive charge from the putative ion permeation path (*Figure 4D*).

Similar to its paralogue mTMEM16A, the interacting α-helices 4 and 6 disengage towards the cytoplasm and open a hydrophilic gap to the membrane, which causes a dilation of the pore at the

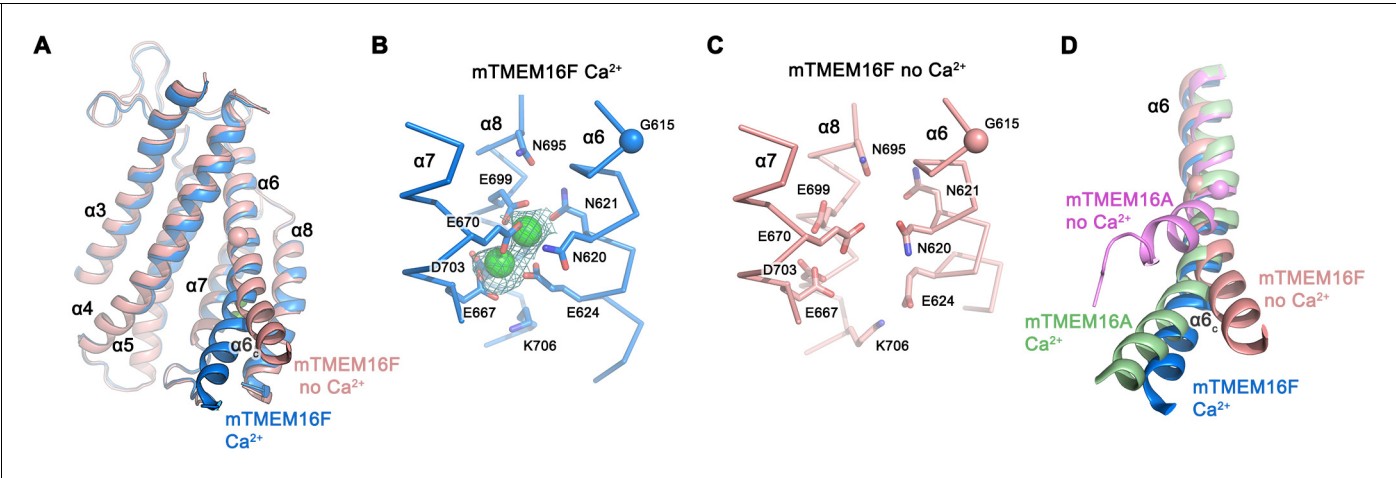

**Figure 5.** Ca²⁺-binding site and conformational changes. (**A**) Superposition of α-helices 3–8 of mTMEM16F in the presence (blue) and absence of Ca²⁺ (coral). (**B** and **C**) Close-up of the Ca²⁺-binding site in complex (**B**) or in absence of Ca²⁺ (**C**). Coordinating residues are depicted as sticks, calcium ions as green spheres (with surrounding cryo-EM density contoured at 5σ). (**D**) Comparison of Ca²⁺-induced movements in mTMEM16F (Ca²⁺-bound in blue, Ca²⁺-free in coral) and mTMEM16A (Ca²⁺-bound (PDBID 5OYB) in green, Ca²⁺-free (PDBID 5OYG) in pink). Shown is a cartoon superposition of α6. A-D. Pivot for movement is depicted as sphere.

DOI: https://doi.org/10.7554/eLife.44365.020

The following figure supplements are available for figure 5:

**Figure supplement 1.** Cryo-EM density of the Ca²⁺-binding site.
DOI: https://doi.org/10.7554/eLife.44365.021

**Figure supplement 2.** Cryo-EM density of a putative Ca²⁺-binding site located at the end of α10.
DOI: https://doi.org/10.7554/eLife.44365.022

intracellular vestibule (*Figure 4D*). In mTMEM16F, this region contains the scrambling domain (SCRD), encompassing the intracellular parts of α4 and α5, which was previously described as determinant for lipid movement (*Gyobu et al., 2017*; *Yu et al., 2015*) (*Figure 4D,E*). The intracellular gap of mTMEM16F also parallels the equivalent region of nhTMEM16, but the closure of the cavity towards the extracellular side interrupts the polar and membrane-accessible furrow found in the Ca²⁺-bound 'open state' of the fungal scramblase nhTMEM16 and instead resembles its Ca²⁺-bound intermediate observed in nanodiscs (*Kalienkova et al., 2019*) (*Figure 3D,E*). The region of the closed cavity that faces the lipid bilayer in mTMEM16F is hydrophobic and resembles mTMEM16A, except for few residues such as Gln 623, located on α6 facing the SCRD, and Lys 370 located at the extracellular domain close to the membrane boundary (*Figure 4D*; *Figure 4—figure supplement 1A*).

## Calcium activation

The structure of mTMEM16F in the presence of Ca²⁺ defines the location of two Ca²⁺ ions in each subunit that are bound to a site which strongly resembles its counterpart in nhTMEM16 (*Brunner et al., 2014*) and mTMEM16A (*Paulino et al., 2017a*) (*Figure 5*; *Figure 5—figure supplement 1A*). In this site, conserved acidic and polar residues located on α-helices 6–8 coordinate the ions with similar geometry. Different from its paralogue functioning as anion channel, mTMEM16F does not contain an insertion of a residue in α6 near the hinge (*Figure 3—figure supplement 1*), which causes a partial unwinding and the formation of a π-bulge in mTMEM16A (*Paulino et al., 2017a*). Consequently, the helix in the ligand-bound conformation of mTMEM16F might be in a less strained conformation.

The conformational changes that take place in mTMEM16F upon ligand binding can be appreciated in the comparison of Ca²⁺-bound and Ca²⁺-free conformations (*Figures 2C* and *5A–C*; *Figure 5—figure supplement 1*). As in mTMEM16A, the absence of Ca²⁺ causes a local transition confined to the pore region, whereas the remainder of the structure remains largely unaffected (*Figures 2C* and *5A*). Besides comparably small rearrangements of α3 and α4, which are less

pronounced in the nanodisc structures (*Figure 2—figure supplement 6*), the largest difference is found at the intracellular half of α6 (α6$_c$, *Figure 5A*). In the presence of Ca$^{2+}$, the bound ligands provide an interaction platform for negatively charged and polar residues on α6, thereby immobilizing the helix in the observed conformation (*Figure 5B*). In the ligand-free state, this helix has altered its conformation in response to the loss of the positively charged interaction partner, leading to an increase in mobility of α6$_c$ as indicated by the weaker density (*Figure 5—figure supplement 1B*). The transition can be described by a rigid-body movement of α6$_c$ by 20°, around a hinge located close to a conserved glycine residue (Gly 615, *Figure 3—figure supplement 1*), whose equivalent in mTMEM16A was shown to be critical for conformational rearrangements (*Paulino et al., 2017a*). However, in contrast to mTMEM16A, the observed change in mTMEM16F is considerably smaller, proceeds in the opposite direction and it is not accompanied by a rotation around the helix axis (*Figure 5D*). Thus, whereas in the apo-form of mTMEM16A α6$_c$ approaches α4, thereby closing the gap between both helices, the movement of α6$_c$ in mTMEM16F is directed away from α4 (*Figure 5A,D*). The observed structural transition opens the access of the ion binding site to the cytoplasm and widens the gap to the SCRD, which likely contributes to the inhibition of scrambling by increasing the energy barrier for lipid flipping by a currently unclear mechanism.

In the recent structure of hTMEM16K, which was crystallized in 100 mM Ca$^{2+}$ (*Bushell et al., 2018*), besides the two Ca$^{2+}$ in the consensus site responsible for activation, an additional Ca$^{2+}$ ion was identified to bind to a site at the intracellular end of α10. This ion is coordinated by negatively charged residues located on α2 and the loop following α10, which are conserved among mammalian but not fungal homologues. Residual density in the Ca$^{2+}$-containing data, which is absent in data of Ca$^{2+}$-free samples, indicate the presence of an ion at the equivalent position in mTMEM16F (*Figure 5—figure supplement 2*). A similar unassigned density is also found in the Ca$^{2+}$-containing data of mTMEM16A (*Paulino et al., 2017b*), but not in fungal homologues (*Falzone et al., 2019*; *Kalienkova et al., 2019*), thus revealing a position of still unclear functional relevance that is shared among mammalian TMEM16 family members and can be occupied by Ca$^{2+}$ at μM concentrations.

## Functional characterization of mutants of the Ca$^{2+}$-binding site

Whereas the described structures revealed the detailed architecture of mTMEM16F and its conformational changes in response to ligand binding, the similarity of the protein to its paralogue working as anion channel leaves important aspects of structure-function relationships ambiguous.

The first question we have addressed was whether Ca$^{2+}$-binding to the conserved site of mTMEM16F would affect scrambling and ion conduction in a similar manner. We have thus investigated the scrambling activity of mutants of the Ca$^{2+}$-binding site and found a strong decrease in the potency of Ca$^{2+}$ in the mutant E667Q located on α7 and little detectable activity in the mutant E624Q on α6 (*Figures 5B* and *6A*; *Figure 6—figure supplement 1A,B*), consistent with a very similar phenotype of equivalent mutants on the activation of the chloride channel mTMEM16A (*Brunner et al., 2014*; *Lim et al., 2016*; *Tien et al., 2014*; *Yu et al., 2012*). By using patch-clamp electrophysiology, we found a comparable effect of the same mutations on the activation of currents, in line with previous observations (*Yang et al., 2012*), thus emphasizing the mutuality in the regulation of both processes (*Figures 5B* and *6B*; *Figure 6—figure supplement 1C,D*). We next investigated the role of the observed conformational change of α6 for activation and the importance of the conserved glycine position as hinge for its movement. In patch-clamp experiments, the mutation of the conserved Gly 615 to alanine increased the potency of Ca$^{2+}$ (*Figure 6B*; *Figure 6—figure supplement 1E*), indicating a stabilization of the open state upon mutation of this flexible position, which parallels a behavior observed in the channel mTMEM16A (*Paulino et al., 2017a*; *Peters et al., 2018*). Hence, the effect of mutations on the Ca$^{2+}$-binding site and the gating hinge indicate that the conformational changes observed in α6 are the basis of a common mechanism of activation of ion conduction and scrambling in mTMEM16F.

## Functional characterization of mutants affecting ion conduction and scrambling

After the characterization of the effect of mutations in the Ca$^{2+}$-binding site on activation, we were interested in other structural elements mediating ion conduction and scrambling. The resemblance of the pore region of mTMEM16F and mTMEM16A suggests that ion conduction could proceed at

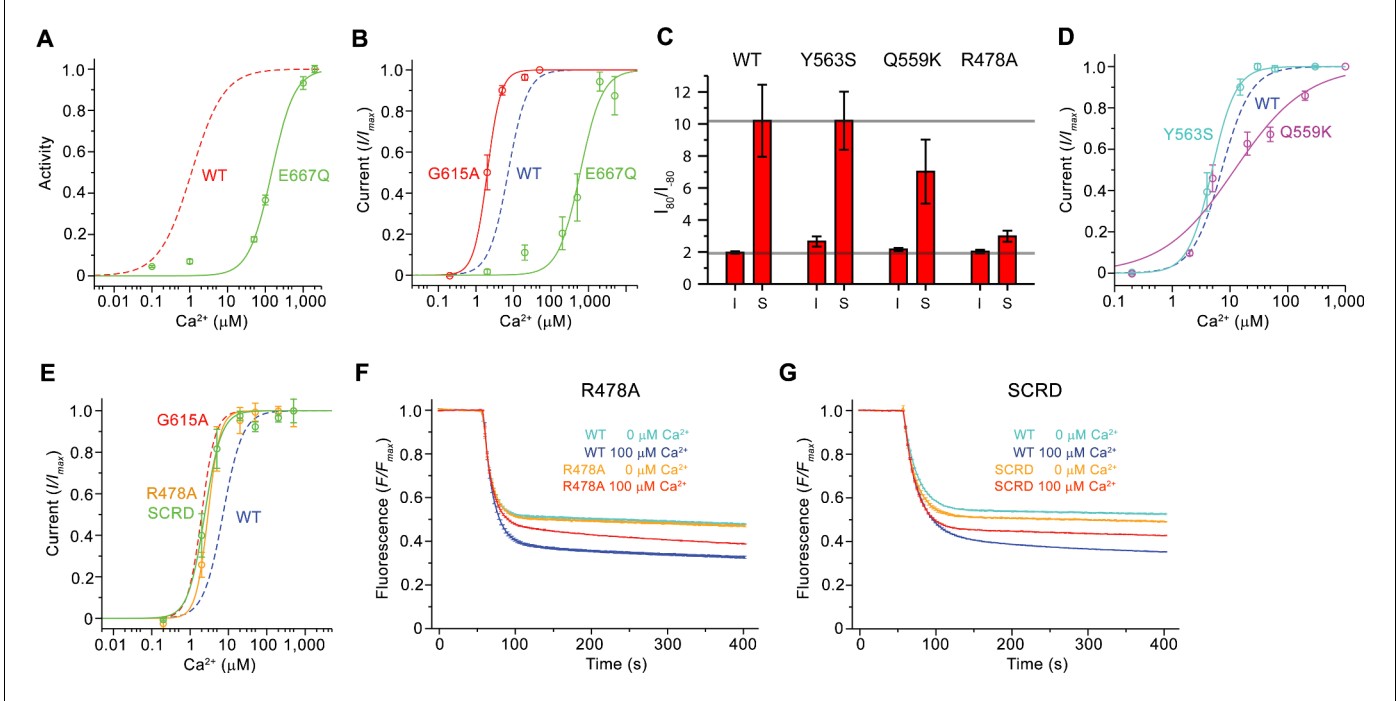

**Figure 6.** Functional properties of mutants. (**A**) $Ca^{2+}$ concentration-response relationship of lipid scrambling in the binding site mutant E667Q. Data show average of either six (0, 1, 10, 100, 1000 µM) or three (other concentrations) independent experiments from two protein reconstitutions. (**B**) $Ca^{2+}$ concentration-response relationship of currents of E667Q (n = 4) and the hinge residue G615A (n = 4). (**A and B**) For comparison WT traces are shown as dashed lines. (**C**) Rectification indices ($I_{80mV}/I_{-80mV}$) of instantaneous (I) and steady-state currents (S) of WT (n = 12) and the mutants Y563S (n = 8), Q559K (n = 10) and R478A (n = 10). Unlike Q599K, the difference between R478A and WT was found to be statistically significant. (**D and E**) $Ca^{2+}$ concentration-response relationship of currents of mutants Y563S (n = 3) and Q559K (n = 5) (**D**) and R478A (n = 5) and mTMEM16F$^{SCRD}$ (SCRD, n = 6) (**E**). (**F and G**) Scrambling of the mutants R478A (**F**) and mTMEM16F$^{SCRD}$ (SCRD) (**G**). F, G Data show averages of three independent experiments of one protein reconstitution. WT reconstituted in a separate set of liposomes from the same batch is shown in comparison. B, D, E Data show averages from independent biological replicates. A-G, Errors are s.e.m.

DOI: https://doi.org/10.7554/eLife.44365.023

The following figure supplements are available for figure 6:

**Figure supplement 1.** Electrophysiology and lipid scrambling data of mutants of the $Ca^{2+}$-binding site and the gating hinge.
DOI: https://doi.org/10.7554/eLife.44365.024

**Figure supplement 2.** Electrophysiology and lipid scrambling data of mutants of the pore region.
DOI: https://doi.org/10.7554/eLife.44365.025

the same location after local conformational changes that increase its diameter. We have thus investigated the influence of mutations of pore-lining residues on ion conduction and gating by analyzing the $Ca^{2+}$ concentration-response relationships, ion selectivity and the rectification properties of currents. In all mutants, we either truncated the residues to alanine or replaced them with their equivalent amino acid found in mTMEM16A. Since, in the cryo-EM structure, Tyr 563 constricts the pore of mTMEM16F (*Figure 4A,C*; *Figure 4—figure supplement 1A,B*), we investigated the mutant Y563S. A similar $Ca^{2+}$ potency and ion selectivity as the WT and no noticeable alteration of the rectification properties of currents were found (*Figure 6C,D*; *Figure 6—figure supplement 2A–C*), thus suggesting that Tyr 563 does not impose a rate-limiting barrier to ion conduction in the open state.

As described previously (*Yang et al., 2012*), the mutation of Gln 559 to lysine at the boundary between the intracellular vestibule and the narrow neck (*Figure 4D*) decreases the weak cation over anion selectivity of mTMEM16F, as expected for the introduction of a positively charged residue interacting with permeating ions (*Figure 6—figure supplement 2D*). In the mutant Q559K, we find a slightly lower $Ca^{2+}$ potency and a markedly decreased slope in the $Ca^{2+}$ concentration-response relationship compared to WT, as well as a moderate decrease in the rectification of steady-state currents, an effect that is particularly pronounced in a subset of the data (*Figure 6C,D*; *Figure 6—*

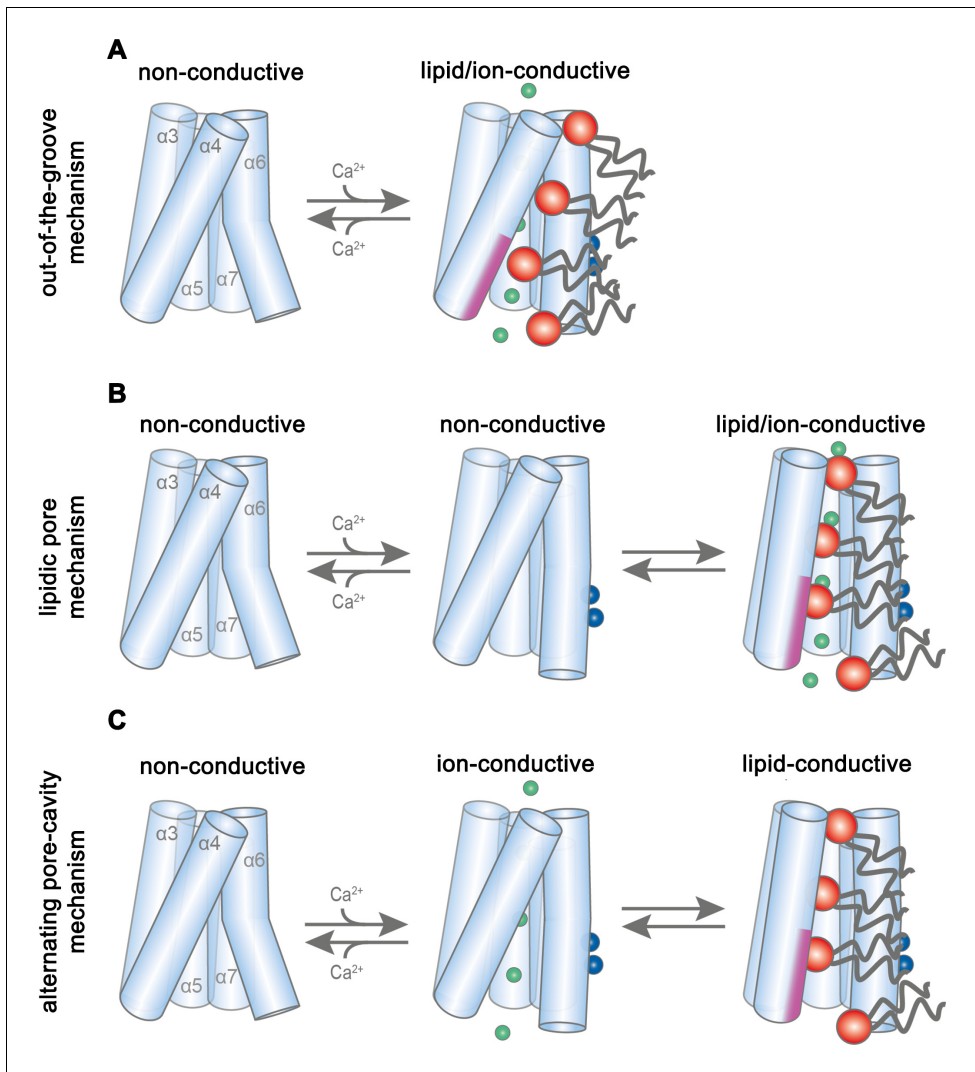

**Figure 7.** Potential transport mechanisms in mTMEM16F. (**A**) 'Out-of-the-groove mechanism': upon calcium-activation the subunit cavity remains closed, ions and lipids permeate separate parts of the protein. (**B**) 'Lipidic pore mechanism': upon $Ca^{2+}$ binding, the cavity opens with ions and lipids permeating the same open cavity conformation. (**C**) 'Alternating pore-cavity mechanism': After $Ca^{2+}$-binding the open lipid-conductive cavity and ion conductive protein-surrounded pore are in equilibrium. This mechanism appears most likely in light of the functional data on mTMEM16F and the results on a related study on the activation of the lipid scramblase nhTMEM16 (*Kalienkova et al., 2019*). A-C, Schematic representation of mTMEM16F α3–7 depicted as cylinders, bound calcium ions as blue spheres, conducting chloride ions as green spheres, scrambling lipid head groups as red spheres and their respective acyl chains as grey lines. The scrambling domain (SCRD) is indicated as purple area at the intracellular side of α-helix 4.

DOI: https://doi.org/10.7554/eLife.44365.026

figure supplement 2A,E). In contrast to the subtle phenotype of Y563S, the mutation of the conserved Arg 478 (*Figure 4B*) at the extracellular entrance of the narrow neck to alanine, yields a several-fold decreased current density compared to WT, yet with increased potency of $Ca^{2+}$ and strongly decreased voltage-dependent relaxations. Consequently, the strong rectification at steady-state disappears, suggesting that the residue plays an important role during channel activation (*Figure 6E*; *Figure 6—figure supplement 2A,F*). Collectively, the observed changes in the activation

and conduction properties of the mutants Q559K and R478A are consistent with their location in the pore. The increase of $Ca^{2+}$-potency found in R478A and the strongly decreased time-dependent reduction of currents at negative voltage might indicate a stabilization of the open state. Moreover, the change in selectivity observed for the mutant Q559K provides direct evidence for its location at the pore lining.

As the mutation R478A at the extracellular part of the pore affected channel properties, we investigated its effect on lipid scrambling. Strikingly, this mutant showed a strongly decreased rate of lipid permeation (*Figure 6F*; *Figure 6—figure supplement 2G*), consistent with data previously obtained from cellular assays (*Gyobu et al., 2017*), whereas the activity of mutants Y563S and Q559K was robust (*Figure 6—figure supplement 2G–I*). Next, we investigated the scrambling activity of the mutants K370A and Q623F, which change the properties of two polar residues facing the outside of the subunit cavity (*Figure 4B*, *Figure 4—figure supplement 1A*). Both residues are located at opposite ends of the otherwise hydrophobic surface of the contact region between α4 and α6. They were selected under the assumption that mutations would interfere with lipid transport in a mechanism that does not rely on the opening of the subunit cavity. Conversely, they would be located on the outside and thus not interfere with lipid scrambling in a mechanism that requires cavity opening. Since both mutants did not show a detectable effect on scrambling, they are likely not located on the lipid permeation path (*Figure 6—figure supplement 2G,J,K*). By contrast, the strong inhibitory effect of the mutation R478A, suggests that an opening of the subunit cavity is necessary to bring this otherwise buried residue into direct contact with permeating lipids.

Finally, we searched for additional mutants with a different effect on ion conduction and lipid scrambling. Similar to previous data from cellular assays (*Yu et al., 2015*), a construct of mTMEM16F where the SCRD was replaced by the equivalent region of mTMEM16A (mTMEM16F$^{SCRD}$, *Figure 4E*) exhibited strongly compromised scrambling activity in our in vitro experiments (*Figure 6G*; *Figure 6—figure supplement 2G*). In contrast, when investigating the TMEM16F$^{SCRD}$ construct by electrophysiology, we observed robust currents, although with a several-fold reduction in current amplitudes, and an increased potency for $Ca^{2+}$ compared to WT (*Figure 6E*; *Figure 6—figure supplement 2L*). Thus, besides the point mutant R478A, also the SCRD chimera appears to exert different effects on lipid and ion permeation, indicating that both processes might be less coupled than previously anticipated (*Jiang et al., 2017*; *Whitlock and Hartzell, 2016*; *Yu et al., 2015*).

## Discussion

By combining data from cryo-electron microscopy, in vitro scrambling assays and electrophysiology, our study has provided insight into the diverse functional behavior of mTMEM16F, a member of the TMEM16 family that facilitates the transport of lipids and ions across cellular membranes. Both processes are controlled by the binding of $Ca^{2+}$ to a site located within the transmembrane domain that is accessible from the cytoplasm and they are mediated by the same structural element located at the periphery of each subunit. Whereas in a cellular environment, the significance of TMEM16F mediated lipid movement is established, the role of ion conduction is currently unclear. Notably, due to the strong outward rectification, ionic currents would be small at resting potential but they could suffice to provide a shunt for the dissipation of charge during the scrambling of anionic lipids, which probably occurs at comparable rates (*Watanabe et al., 2018*; *Yang et al., 2012*). Although our results offer detailed insights into potential transport mechanisms, there are still open questions related to the uncertain correspondence of the $Ca^{2+}$-bound conformation of the protein to a defined functional state. We thus envision three alternative scenarios for ion and lipid permeation. In all scenarios, the structures obtained in $Ca^{2+}$-free conditions describe an inactive conformation of the protein, while the $Ca^{2+}$-bound structures either represent an intermediate or final state of activation. Whereas, in light of our cryo-EM structures, all three scenarios appear plausible, the combination with functional data points towards a mechanism where ion conduction and scrambling are mediated by distinct protein conformations.

In the first scenario ('out-of-the-groove mechanism'), the here reported $Ca^{2+}$-bound mTMEM16F structures represent a conformation close to the endpoint of the protein activation (*Figure 7A*). Since, in the $Ca^{2+}$-bound state, the extracellular part of the subunit cavity remains shielded from the membrane, the diffusion of lipids would proceed outside of the cavity for about two-thirds of their transition. In this scenario, scrambling would be facilitated by interactions with the SCRD, which

exposes polar residues to the inner leaflet of the bilayer at the gap between α4 and α6. Protein interactions with the lipid headgroups destabilize the membrane and consequently lower the barrier for lipid movement, which does not proceed in tight contact with the protein, as proposed in a recent study (*Malvezzi et al., 2018*). By contrast, ion conduction takes place in a sufficiently expanded protein-enclosed pore separated from the scrambling path. Upon activation, the observed movement of α6 towards α4 would change the structure in vicinity of the SCRD sufficiently to allow both lipid movement and ion conduction. This scenario would imply that there are two classes of TMEM16 scramblases acting by distinct mechanisms. The first class would allow the passage of lipids through a membrane-accessible furrow and include fungal scramblases (*Brunner et al., 2014*; *Falzone et al., 2019*; *Kalienkova et al., 2019*) and their mammalian orthologues residing in intracellular organelles, such as hTMEM16K (*Bushell et al., 2018*). A second class, defined by mTMEM16F and its orthologues, which are located in the plasma membrane, are closely related to the ion channels mTMEM16A and B and thus show pronounced ion conduction properties but do not contain a membrane-accessible polar subunit cavity.

In a second scenario, scrambling would proceed on the surface of a subunit cavity that has opened to the membrane (*Figure 7B,C*). In this case, the Ca²⁺-bound mTMEM16F structures display intermediates, whereby the conformational heterogeneity of α3 and α4 observed in the nanodisc samples indicates a potential rearrangement towards a fully active protein conformation (*Figure 2—figure supplements 3I* and *4I*). Consequently, the comparison to Ca²⁺-free structures reveals the first ligand-dependent activation step. Full activation of the protein to a scrambling-competent state requires a consecutive transition that might be facilitated by lipid interactions with the protein at the SCRD, thereby facilitating the conformational change of α4 (*Lee et al., 2018*). This transition would lead to the exposure of a membrane-spanning hydrophilic furrow as observed in the scramblase nhTMEM16 (*Brunner et al., 2016*). This scenario is generally consistent with the activation mechanism for the fungal scramblases nhTMEM16 (*Kalienkova et al., 2019*) and afMEM16 (*Falzone et al., 2019*) and the human scramblase TMEM16K (*Bushell et al., 2018*). The fully active conformation was not observed in our structures, since it is either transient, as proposed for lipid scrambling in rhodopsin (*Morra et al., 2018*), or disfavored in our lipid nanodiscs, potentially due to an impairing lipid composition or the absence of interacting components (*Schreiber et al., 2018*; *Ye et al., 2018*). In such a scenario, we foresee two potential alternatives for the observed ion conduction. Conduction could either proceed in the same conformation as scrambling, in a pore that is partially lined by translocating lipids and thus be directly coupled to lipid movement ('lipidic pore mechanism'), as suggested in previous investigations (*Jiang et al., 2017*; *Whitlock and Hartzell, 2016*) (*Figure 7B*). Alternatively, ion conduction and scrambling could be mediated by distinct alternating conformations, which are at equilibrium in the Ca²⁺-bound state (*Falzone et al., 2018*; *Kalienkova et al., 2019*). Here, scrambling proceeds in an open subunit cavity, as observed for nhTMEM16, afTMEM16 and hTMEM16K, while ion conduction is catalyzed by a protein-enclosed pore that resembles the Ca²⁺-bound mTMEM16F and mTMEM16A structures ('alternating pore-cavity mechanism', *Figure 7C*).

Whereas our structures do not show an opened subunit cavity, the functional data are most consistent with distinct conformations facilitating ion conduction and scrambling, as described in the 'alternating pore-cavity mechanism'. In this case, mutations could shift the equilibrium between both states and have opposite effects on either function. This is observed for the mutants R478A and TMEM16F^SCRD, which both mediate Ca²⁺-activated currents with smaller magnitude but increased ligand potency compared to WT, while both show impaired lipid permeation properties (*Figure 6E–G*). Particularly, this scenario would account for the strong effect of the mutation of Arg 478 on lipid permeation (*Figure 6F*) (*Gyobu et al., 2017*), which in our structures is buried in the protein but would become exposed to the membrane in an open cavity. In contrast, the mutation of polar residues on the outside of the subunit cavity, which would contribute to scrambling in the 'out-of-the-groove mechanism', have little impact on lipid movement (*Figure 6—figure supplement 2J,K*). A shift in the equilibrium of the two states might also underlie the decrease in the open probability of the ion conduction pore at negative voltages (*Figure 1C*, *Figure 1—figure supplement 1G*). A definitive resolution of the mechanistic ambiguity will require further investigations, for which our study has provided an important foundation.

# Materials and methods

## Key resources table

| Reagent type (species) or resource | Designation | Source or reference | Identifier | Additional information |
|---|---|---|---|---|
| Antibody | Mouse monoclonal Anti-c-Myc | Millipore Sigma | Cat#M4439; Clone#9E10 | Dilution – 1:5000 |
| Antibody | Peroxidase Affinipure goat anti-mouse IgG | Jackson Immunoresearch | Cat#115-035-146 | Dilution – 1:10000 |
| Chemical compound, drug | FuGENE six transfection reagent | Promega | Cat# E2691 | |
| Chemical compound, drug | Hygromycin B | Gibco, Thermo Fisher Scientific | Cat#10687010 | |
| Chemical compound, drug | Tetracycline hydrochloride | Millipore Sigma | Cat#T7660 | |
| Chemical compound, drug | $n$-dodecyl-β-d-maltopyranoside, Solgrade | Anatrace | Cat#D310S | |
| Chemical compound, drug | EX-CELL 293 Serum-Free medium | Millipore Sigma | Cat#14571C | |
| Chemical compound, drug | Fetal bovine serum | Millipore Sigma | Cat#F7524 | |
| Chemical compound, drug | L-glutamine | Millipore Sigma | Cat#G7513 | |
| Chemical compound, drug | Penicillin-streptomycin | Millipore Sigma | Cat#P0781 | |
| Chemical compound, drug | Valproic acid | Millipore Sigma | Cat#P4543 | |
| Chemical compound, drug | HyClone HyCell TransFx-H medium | GE Healthcare | Cat#SH30939.02 | |
| Chemical compound, drug | Poloxamer 188 | Millipore Sigma | Cat#P5556 | |
| Chemical compound, drug | Polyethylenimine MAX 40 K | Polysciences | Cat# 24765–1 | |
| Chemical compound, drug | Dulbecco's Modified Eagle's Medium - high glucose | Millipore Sigma | D5671 | |
| Chemical compound, drug | Dulbecco's Phosphate Buffered Saline | Millipore Sigma | D8537 | |
| Chemical compound, drug | Digitonin Reagent USP | PanReac AppliChem | Cat#A1905 | |
| Chemical compound, drug | Calcium nitrate tetrahydrate | Millipore Sigma | Cat#C4955 | |
| Chemical compound, drug | Sodium chloride | Millipore Sigma | Cat#71380 | |
| Chemical compound, drug | HEPES | Millipore Sigma | Cat#H3375 | |
| Chemical compound, drug | Ethylene glycol-bis(2-aminoethylether)-N,N,N′,N′-tetraacetic acid | Millipore Sigma | Cat#03777 | |
| Chemical compound, drug | cOmplete, EDTA-free Protease Inhibitor Cocktail | Roche | Cat#5056489001 | |
| Chemical compound, drug | Digitonin, High Purity - Calbiochem | EMD Millipore | Cat#300410 | |

*Continued on next page*

*Continued*

| Reagent type (species) or resource | Designation | Source or reference | Identifier | Additional information |
|---|---|---|---|---|
| Chemical compound, drug | Biotin | Millipore Sigma | Cat#B4501 | |
| Recombinant protein | PNGase F | Raimund Dutzler laboratory | NA | |
| Commercial assay or kit | Amicon Ultra-4–100 KDa cutoff | EMD Millipore | Cat#UFC8100 | |
| Commercial assay or kit | 0.22 µm Ultrafree-MC Centrifugal Filter | EMD Millipore | Cat#UFC30GV | |
| Chemical compound, drug | 1-palmitoyl-2-oleoyl-glycero-3-phosphocholine | Avanti Polar Lipids, Inc | Cat#850457C | |
| Chemical compound, drug | 1-palmitoyl-2-oleoyl-sn-glycero-3-phospho-(1'-rac-glycerol) | Avanti Polar Lipids, Inc | Cat#840457C | |
| Chemical compound | Diethyl ether | Millipore Sigma | Cat#296082 | |
| Commercial assay or kit | Bio-Beads SM-2 Adsorbents | Bio-Rad | Cat# 1523920 | |
| Chemical compound, drug | Soybean Polar Lipid Extract | Avanti Polar Lipids, Inc | Cat#541602C | |
| Chemical compound, drug | Cholesterol | Millipore Sigma | Cat#C8667 | |
| Chemical compound | 18:1-06:0 NBD-PE | Avanti Polar Lipids, Inc | Cat#810155C | |
| Chemical compound, drug | 18:1-06:0 NBD-PS | Avanti Polar Lipids, Inc | Cat#810194C | |
| Chemical compound, drug | 14:0 NBD-PE | Avanti Polar Lipids, Inc | Cat#810143 C | |
| Chemical compound, drug | Potassium chloride | Millipore Sigma | Cat#746436 | |
| Chemical compound, drug | Triton X-100 | Millipore Sigma | Cat#T9284 | |
| Chemical compound, drug | Sodium dithionite | Millipore Sigma | Cat#157953 | |
| Chemical compound, drug | N-Methyl-D-glucamine | Millipore Sigma | Cat#66930 | |
| Chemical compound, drug | Sulphuric acid 95–97% | EMD Millipore | Cat# 1.00731.1000 | |
| Commercial assay or kit | Borosilicate glass capilliary with filament | Sutter Instrument | Cat#BF150-86-10HP | |
| Commercial assay or kit | Pierce Streptavidin Plus UltraLink Resin | Thermo Fisher Scientific | Cat#53117 | |
| Commercial assay or kit | Superose 6 10/300 GL | GE Healthcare | Cat#17-5172-01 | |
| Commercial assay or kit | Flp-In T-Rex Core Kit | ThermoFisher Scientific | Cat#K650001 | |
| Commercial assay or kit | Whatman Nuclepore Track-Etched Membranes diam.19mm, pore size 0.4µm, polycarbonate | Millipore Sigma | Cat#WHA800282 | |
| Commercial assay or kit | Microforge | Narishige | NA | |
| Commercial assay or kit | Axopatch 200B amplifer | Molecular Devices | NA | |

*Continued on next page*

*Continued*

| Reagent type (species) or resource | Designation | Source or reference | Identifier | Additional information |
|---|---|---|---|---|
| Commercial assay or kit | Digidata 1440 | Molecular Devices | NA | |
| Commercial assay or kit | 300 mesh Au 1.2/1.3 cryo-EM grids | Quantifoil | Cat#N1-C14nAu30-01 | |
| Cell line (Human) | Flp-In T-REx 293 | ThermoFisher Scientific | Cat#R78007 | |
| Cell line (Human) | HEK-293T | ATCC | Cat#CRL-1573 | |
| Cell line (Human) | HEK293S GnTI⁻ | ATCC | Cat#CRL-3022 | |
| Recombinant DNA | Mouse mTMEM16F open reading frame | Dharmacon - Horizon Discovery | GenBank#BC060732 | |
| Recombinant DNA | Mammalian expression vector with C-terminal 3C protease cleavage site, Myc tag and streptavidin binding peptide | Raimund Dutzler laboratory | NA | |
| Recombinant DNA | Mammalian expression vector with C-terminal 3C protease cleavage site, Venus and Myc tags and streptavidin binding peptide | Raimund Dutzler laboratory | NA | |
| Recombinant DNA | Synthesized mTMEM16F$^{SCRD}$ cDNA | GenScript | NA | |
| Recombinant DNA | Membrane scaffold protein (MSP) 2N2 | Stephen Sligar laboratory | Addgene:Cat#29520 | |
| Software, algorithm | WEBMAXC calculator | *Bers et al., 2010* | http://maxchelator.stanford.edu/webmaxc/webmaxcS.htm | |
| Software, algorithm | Axon Clampfit 10.7 | Molecular Devices | NA | |
| Software, algorithm | Axon Clampex 10.6 | Molecular Devices | NA | |
| Software, algorithm | Focus 1.1.0 | *Biyani et al. (2017)* | https://focus.c-cina.unibas.ch/about.php | |
| Software, algorithm | MotionCorr2 1.1.0 | *Zheng et al. (2017)* | http://msg.ucsf.edu/em/software/motioncor2.html | |
| Software, algorithm | CTFFIND 4.1 | *Rohou and Grigorieff (2015)* | http://grigoriefflab.janelia.org/ctf | |
| Software, algorithm | Relion v 2.1 and 3.0 | *Kimanius et al., 2016 Zivanov et al., 2018* | https://www2.mrc-lmb.cam.ac.uk/relion/ | |
| Software, algorithm | Phenix 1.13 | *Adams et al. (2010)* | http://http://phenix-online.org/ | |
| Software, algorithm | Coot 0.8.9.1 | *Emsley and Cowtan (2004)* | https://www2.mrc-lmb.cam.ac.uk/personal/pemsley/coot/ | |
| Software, algorithm | Pymol 2.0 | Schrodinger LLC | https://pymol.org/2/ | |
| Software, algorithm | Chimera 1.12 | *Pettersen et al. (2004)* | https://www.cgl.ucsf.edu/chimera/ | |

## Cell lines

Flp-In T-REx 293 cell lines stably expressing mTMEM16F were adapted to suspension cultures and were grown at 37°C and 5% $CO_2$ in EX-CELL 293 Serum-free medium supplemented with 1% fetal bovine serum, 6 mM L-glutamine and 100 U/ml penicillin–streptomycin. GnTI⁻ cells were grown in HyClone HyCell TransFx-H medium supplemented with 1% fetal bovine serum, 4 mM L-glutamine, 1 g/l poloxamer 188 and 100 U/ml penicillin–streptomycin. Adherent HEK293T cells were grown in DMEM medium supplemented with 10% fetal bovine serum, 2 mM L-glutamine, 1 mM Sodium

pyruvate and 100 U/ml penicillin–streptomycin. Mycoplasma test performed on the used cells was negative.

## Construct preparation

The gene encoding mTMEM16F (GenBank: BC060732) was obtained from Dharmacon-Horizon Discovery and stably inserted into a tetracycline-inducible HEK293T cell line using the Flp-In T-Rex System. For that purpose, the mTMEM16F sequence was cloned into two pcDNA5/FRT expression vectors, one with C-terminal Rhinovirus 3C protease recognition site, Myc and Streptavidin-binding peptide (SBP) tags and the other with an extra Venus tag before the Myc and SBP tags. Both cell lines were created in the same manner: briefly, 1 μg of mTMEM16F expression vector and 9 μg pOG44 recombinase were co-transfected after mixing with 30 μg Fugene6 transfection reagent. Hygromycin (50 μg/ml in the first week and 100 μg/ml afterwards) was used as selection marker and resistant foci appeared after 2 weeks. Per cell line, 15 foci were expanded and mTMEM16F expression was assayed after tetracycline induction (2 μg/ml) for 48 hr, extraction with 2% $n$-dodecyl-β-d-maltopyranoside (DDM) and Western blot analysis using a mouse monoclonal anti-Myc and a Peroxidase Affinipure goat anti-mouse IgG antibodies (*Figure 1—figure supplement 1A*).

For mutagenesis, the sequence of mTMEM16F was cloned into FX-cloning compatible pcDNA3.1 vectors (*Geertsma and Dutzler, 2011*) and point mutations were introduced with the non-overlapping primers modified QuikChange method (*Zheng et al., 2004*). The sequence was not codon optimized except for the removal of a *SapI* restriction site. For experiments requiring protein purification, the sequence was cloned into an expression vector with C-terminal 3C recognition site, Myc and SBP tags. For electrophysiology experiments, an expression vector with either a Venus (in the case of WT and mutants E624Q, E667Q and Y563S) or a GFP tag (in the case of mutants Q559K, G615A, R478A and mTMEM16F$^{SCRD}$) between the 3C site and Myc-SBP was used instead. All constructs were verified by sequencing. The mTMEM16F$^{SCRD}$ construct, with residues 525–559 mutated to their equivalent in mTMEM16A, was synthesized by GenScript. The plasmid used for the expression of the membrane scaffold protein (MSP) 2N2 was obtained from Addgene (plasmid #29520).

## Protein expression and purification

Wild-type mTMEM16F was expressed by tetracycline induction (2 μg/ml) of the stably transfected mTMEM16F-3C-Myc-SBP cell line for 48–70 hr. After induction, the medium was further supplemented with 3.5 mM valproic acid. Mutant mTMEM16F constructs were expressed by transient transfection of GnTI⁻ cells. For cell transfection, DNA was complexed in a 1:2.5 ratio with Polyethylenimine MAX 40 K in non-supplemented DMEM medium for 20 min before addition to the cells. After transfection, the medium was further supplemented with 3.5 mM valproic acid and the cells were collected after 48–60 hr, washed with PBS and stored at −80°C until further use.

Protein purification of wild-type and mutant mTMEM16F was carried out at 4°C and was completed within 14 hr. In all cases, we have purified the protein under Ca$^{2+}$-free conditions and added 1 mM Ca$^{2+}$ when indicated during cryo-EM sample preparation. Cells were resuspended in 2% digitonin (PanReac AppliChem), 150 mM NaCl, 20 mM HEPES, pH 7.5, 5 mM EGTA and protease inhibitors and the membranes were solubilized by gentle mixing for 2 hr. Solubilized proteins were isolated by centrifugation at 85 000 $g$ for 30 min and allowed to bind to streptavidin UltraLink resin beads for 2 hr under gentle agitation. The beads were loaded onto a gravity column and washed with 60 column volumes (CV) of SEC buffer containing 0.1% digitonin (EMD Millipore), 150 mM NaCl, 20 mM HEPES, pH 7.5 and 2 mM EGTA. Bound protein was eluted with 3 CV of SEC buffer supplemented with 4 mM biotin. Samples used for cryo-EM were digested with PNGaseF for 2 hr. Protein was concentrated using a 100 kDa cutoff filter, filtered through a 0.22 μm filter and loaded onto a Superose 6 10/300 GL column pre-equilibrated with SEC buffer. Protein-containing peak fractions were pooled, concentrated, filtered and immediately used for either cryo-EM sample preparation or reconstitution into nanodiscs or liposomes. The MSP 2N2 protein was expressed and purified as described (*Ritchie et al., 2009*). In this case, the N-terminal poly-histidine tag was not cleaved after purification.

Nanodisc reconstitution was performed as described (*Gao et al., 2016*) with minor modifications. Purified protein was incorporated into nanodiscs composed of a1-palmitoyl-2-oleoyl-glycero-3-phosphocholine (POPC) and 1-palmitoyl-2-oleoyl-sn-glycero-3-phospho-(1'-rac-glycerol) (POPG) mixture

at a molar ratio of 3:1. After mixing, the chloroform-dissolved lipids were initially dried under a nitrogen stream, washed with diethyl ether and again dried under nitrogen stream and vacuum desiccation overnight. The lipid mix was solubilized in 30 mM DDM, at a final lipid concentration of 10 mM. A molar ratio of 2:10:2200 of mTMEM16F:MSP 2N2:lipids was used for reconstitution. Purified protein was incubated with the lipid mix on ice for 40 min. Subsequently, 2N2 was added to the mixture and further incubated for 40 min on ice. Detergent was removed overnight after addition of 50 mg of SM-II biobeads per mg of DDM. Biobeads were removed from the clear sample by column filtration and the nanodisc suspension was incubated with streptavidin UltraLink resin for 2 hr. Further purification proceeded as described for the digitonin-solubilized protein, except that no detergent was present in any of the buffers.

## Liposome reconstitution and scrambling experiments

For the characterization of lipid scrambling, purified mTMEM16F was reconstituted into liposomes containing trace amounts (0.5% weight/weight) of fluorescently (nitrobenzoxadiazole, NBD) labeled lipids. The lipid mix used for the majority of scrambling experiments was soybean polar lipids extract with 20% cholesterol (mol/mol) and 0.5% 18:1-06:0 NBD-PE. To investigate whether scrambling proceeds independently of the chemical nature of the headgroup of the labeled lipid or the location of the fluorophore, the tail labeled 18:1-06:0 NBD-PE was substituted by 18:1-06:0 NBD-PS or head-labeled 14:0 NBD-PE while maintaining the other liposome components unchanged. To characterize the protein activity in the lipid mix used for the preparation of nanodisc samples, we reconstituted mTMEM16F into liposomes composed of 3 POPC:1 POPG: 0.5% 18:1-06:0 NBD-PE. Lipids were treated similarly as the ones used for nanodisc reconstitution until solubilization. Liposomes were prepared as described (*Geertsma et al., 2008*). For liposomes, the lipids were solubilized in 20 mM HEPES pH 7.5, 300 mM KCl and 2 mM EGTA (buffer A) in a final concentration of 20 mg/ml, sonicated, subjected to three freeze-thaw cycles in liquid $N_2$ and stored at $-80°C$. The liposomes were protected from direct light as much as possible. Prior to reconstitution, the lipids were extruded 21 times through a 400 nm pore polycarbonate membrane and diluted to 4 mg/ml in buffer A. Liposomes were destabilized by addition of Triton X-100 aliquots in 0.02% steps until the onset of lipid solubilization monitored by the decrease of the scattering at 540 nm in a spectrophotometer. After addition of another aliquot of 0.1% Triton X-100, the protein was added at a lipid-to-protein ratio of 100:1 (weight/weight) unless stated otherwise to the destabilized liposomes and the mixture was incubated at room temperature (RT) for 15 min under gentle agitation. Subsequently, 20 mg of SM-II biobeads per mg of lipids were added four times to completely remove digitonin and Triton X-100. After 45 min, the sample was moved to 4°C. Proteoliposomes were harvested after 24 hr by filtration to remove the biobeads followed by centrifugation at 150,000 *g* for 30 min. The pelleted proteoliposomes were resuspended in buffer A with solutions containing the appropriate amount of $Ca(NO_3)_2$ to obtain the indicated concentrations of free $Ca^{2+}$ calculated with the online WEBMAXC calculator (*Bers et al., 2010*). The liposomes were resuspended at either 10 or 20 mg/ml and subjected to three freeze-thaw cycles in liquid $N_2$. All remaining steps were carried out at RT. After extrusion through a 400 nm membrane pre-equilibrated in buffer A containing the desired amount of free $Ca^{2+}$, proteoliposomes were diluted to 0.2 mg/ml in 80 mM HEPES pH 7.5, 300 mM KCl, 2 mM EGTA plus $Ca^{2+}$ (buffer B). Scrambling was performed essentially as described for other scramblases of the TMEM16 family (*Brunner et al., 2014*; *Malvezzi et al., 2013*). Scrambling was monitored by spectrofluorimeter measurements of the NBD fluorophore with an excitation wavelength of 470 nm and emission wavelength of 530 nm. After 60 s of recording, 30 mM of the membrane-impermeable reducing agent sodium dithionite was added to irreversibly bleach exposed NBD groups. The total recording time was of 400 s unless stated otherwise and the sample was stirred during the entire measurement. The NBD-fluorescence decay was plotted as $F/F_{max}$. To investigate the ligand-dependence of the scrambling rate, we measured the NBD-fluorescence value 120 s after the addition of dithionite at different $Ca^{2+}$ concentrations and quantified scrambling activity as $1-F_{Ca2+}/F_{0Ca2+}$ since we were unable to detect any pronounced scrambling activity for mTMEM16F in the absence of $Ca^{2+}$. The effect of $Ca^{2+}$ on the scrambling rate was studied using symmetric buffer conditions, with the same $Ca^{2+}$ concentration inside and outside the liposomes. To investigate the delay time between calcium addition and scrambling activation we reconstituted mTMEM16F in calcium-free liposomes and recorded the NBD-fluorescence decay upon addition of 100 μM $Ca(NO_3)_2$ to the outside buffer, 120 s after dithionite addition, and monitored fluorescence for extra 420 s.

The same batch of liposomes was incubated in buffer B containing 100 μM Ca(NO$_3$)$_2$ for 10 min prior to measurement as control.

In empty liposomes, addition of dithionite causes a decrease of the fluorescence to about half of its initial value, due to the bleaching of the fraction of fluorescent lipids located in the outer leaflet of the bilayer (*Figure 1—figure supplement 1C*). In contrast, proteoliposomes containing an active scramblase show a stronger time-dependent decrease of the fluorescence to a plateau, which corresponds to the fraction of fluorescent lipids that are not accessible to dithionite since they either reside in liposomes not containing a reconstituted scramblase or they are located on the inside of a multi-lamellar vesicle (*Malvezzi et al., 2018*; *Ploier and Menon, 2016*) (*Figure 1A*; *Figure 1—figure supplement 1C–F*). For different reconstitutions, we find similar plateau values for empty liposomes and proteoliposomes in Ca$^{2+}$-free conditions. In contrast, the plateau of fully activated samples is consistent for different constructs using the same batch of destabilized lipids but it varies between different reconstitutions, with fluorescence levels ranging from 18% to 30%. We attribute this difference to the efficiency of reconstitution that is dependent on the batch of destabilized liposomes. Independent of the plateau, all reconstitutions showed a very similar Ca$^{2+}$-dependence. Due to the variable reconstitution efficiency, all the mutants were purified in parallel with wild type (WT) mTMEM16F and reconstituted using the same batch of lipids in equivalent reconstitution conditions. Protein incorporation into liposomes for WT and mutants was verified by Western blot against the Myc tag after solubilization of proteoliposomes with DDM. With the exception of mTMEM16F$^{SCRD}$, all mutants reconstituted with an efficiency similar to WT (*Figure 6—figure supplement 2G*). mTMEM16F$^{SCRD}$ behaved well during purification but reconstituted with somewhat lower efficiency. For that purpose, it was compared to WT reconstituted at lower lipid to protein ratio of 150:1. For this case proteoliposomes contained similar protein levels (*Figure 6—figure supplement 2G*).

## Electrophysiology

For electrophysiology, HEK293T cells were transfected with plasmids at a concentration of 8 μg DNA per 100 mm culture dish with 25 μl FuGENE 6 transfection reagent. Transfected cells were identified by Venus or GFP fluorescence and used for patch clamp experiments within 24–72 hr of transfection. WT currents were also recorded from cells stably expressing mTMEM16F without fluorescent fusion protein used for purification. All recordings were performed in the inside-out configuration (*Hamill et al., 1981*) at RT (20–22°C). Inside-out patches were excised from HEK293T cells expressing the mTMEM16F construct of interest after the formation of a gigaohm seal. Seal resistance was typically 4–8 GΩ or higher. Patch pipettes were pulled from borosilicate glass capillaries (OD 1.5 mm, ID 0.86 mm) and were fire-polished using a microforge. Pipette resistance was typically 3–8 MΩ when filled with pipette solutions. Voltage-clamp recordings were performed using the Axopatch 200B amplifier controlled by the Clampex 10.6 software through Digidata 1440. The data were sampled at 10 kHz and filtered at 1 kHz. Solution exchange was achieved using a double-barreled theta glass pipette mounted on an ultra-high speed piezo-driven stepper (Siskiyou). Liquid junction potential was not corrected. Step-like solution exchange was elicited by analogue voltage signals delivered through Digidata 1440. All recordings were measured in symmetrical NaCl solutions except for selectivity experiments. The background current was recorded in Ca$^{2+}$-free solution and subtracted prior to analysis. Ca$^{2+}$-free intracellular solution contained 150 mM NaCl, 5 mM EGTA, and 10 mM HEPES, pH 7.40. High intracellular Ca$^{2+}$ solution, with a free concentration of 1 mM, contained 150 mM NaCl, 5.99 mM Ca(OH)$_2$, 5 mM EGTA, and 10 mM HEPES, pH 7.40. The pH was adjusted using 1 M NMDG-OH solution. Intermediate Ca$^{2+}$ solutions were obtained by mixing high Ca$^{2+}$ and Ca$^{2+}$-free intracellular solutions at the ratio calculated according to the WEBMAXC calculator (*Bers et al., 2010*), free Ca$^{2+}$ concentrations above 1 mM (up to 10 mM) were adjusted by adding CaCl$_2$ from a 1 M stock solution. The Ca$^{2+}$-free intracellular solution was also used as the pipette solution. For permeability experiments using (NMDG)$_2$SO$_4$ to compensate for the ionic strength, the appropriate NaCl concentration was adjusted by mixing NaCl stock solutions and (NMDG)$_2$SO$_4$ stock solutions at the required ratios. Stock NaCl solutions were the same as above. Stock (NMDG)$_2$SO$_4$ solutions contained 100 mM (NMDG)$_2$SO$_4$, 5.99 mM Ca(OH)$_2$, 5 mM EGTA, 10 mM HEPES, pH 7.40, and 100 mM (NMDG)$_2$SO$_4$, 5 mM EGTA, and 10 mM HEPES, pH 7.40.

Current of mTMEM16F runs down in the excised patch configuration. In order to obtain a more accurate EC$_{50}$ of Ca$^{2+}$-activation, rundown correction was performed using a previously described method (*Lim et al., 2016*). In short, a reference Ca$^{2+}$ pulse was applied at a regular time interval

before and after the test pulse. The magnitude of the test pulse was normalized to the average magnitude of the pre- and post-reference pulses. The normalized concentration–response data were fitted to the Hill equation. To correct for rundown during rectification experiments, the instantaneous and steady-state current at each voltage step was divided by the ratio of the remaining current at the 80 mV pre-pulse and expressed as normalized current ($I/I_{80mV}$). The rectification indices of instantaneous currents were calculated as the ratio between the current measured at 80 mV, 100 ms before the voltage jump, and 6 ms after changing to −80 mV at which the capacitive transients have already decayed. For calculation of the rectification index of steady-state currents, the value at −80 mV was taken once the current response has reached a plateau (typically 150–200 ms after the voltage change).

The significance of differences between the constructs was determined with a one-way ANOVA and a Tukey-Kramer post-hoc test. Values were considered significantly different if $p<0.05$. By this criterion, shifts in the $EC_{50}$ for mutants E667Q, G615A and TMEM16F$^{SCRD}$ compared to WT were determined as significant whereas the shift in R478A is slightly above the significance threshold. For the characterization of rectification properties of steady-state currents, only the shift of the mutant R478A compared to WT was calculated to be statistically significant. In case of the mutant Q559K, the high variance due to the presence of two populations of data with strong and weak rectification properties rendered the shift in its value as statistically insignificant.

## Cryo-electron microscopy sample preparation and imaging

2.5 µl of freshly purified protein at a concentration of 3.3 mg ml$^{-1}$, when solubilized in digitonin, and 1 mg ml$^{-1}$, when reconstituted in nanodiscs, were applied on holey-carbon cryo-EM grids (Quantifoil Au R1.2/1.3, 200, 300 and 400 mesh), which were prior glow-discharged at 5 mA for 30 s. For the datasets obtained for the Ca$^{2+}$-bound structures, samples were supplemented with 1 mM CaCl$_2$ before freezing. Grids were blotted for 2–5 s in a Vitrobot (Mark IV, Thermo Fisher) at 10–15°C temperature and 100% humidity, subsequently plunge-frozen in liquid ethane and stored in liquid nitrogen until further use. Cryo-EM data were collected on a 200 keV Talos Arctica microscope (Thermo Fisher) using a post-column energy filter (Gatan) in zero-loss mode, a 20 eV slit, a 100 µm objective aperture, in an automated fashion using EPU software (Thermo Fisher) on a K2 summit detector (Gatan) in counting mode. Cryo-EM images were acquired at a pixel size of 1.012 Å (calibrated magnification of 49,407x), a defocus range from –0.5 to –2 µm, an exposure time of 9 s with a sub-frame exposure time of 150 ms (60 frames), and a total electron exposure on the specimen of about 52 electrons per Å$^2$. Best regions on the grid were manually screened and selected with the help of an in-house script to calculate the ice thickness (manuscript in preparation). Data quality was monitored on the fly using the software FOCUS (*Biyani et al., 2017*).

## Image processing

For the detergent dataset collected in presence of Ca$^{2+}$, the 5633 dose-fractionated cryo-EM images recorded (final pixel size 1.012 Å) were subjected to motion-correction and dose-weighting of frames by MotionCor2 (*Zheng et al., 2017*). The CTF parameters were estimated on the movie frames by ctffind4.1 (*Rohou and Grigorieff, 2015*). Bad images showing contamination, a defocus above –0.5 or below –2 µm, or a bad CTF estimation, were discarded. The resulting 4225 images were used for further analysis with the software package RELION2.1 (*Kimanius et al., 2016*). Particles were initially picked automatically from a subset of the dataset using 2D-class averages from the previously obtained mTMEM16A cryo-EM map (*Paulino et al., 2017a*) as reference. The resulting particles where used to create a better reference for autopicking, which was then repeated on the whole dataset. The final round of autopicking yielded 1,348,247 particles, which were extracted with a box size of 220 pixels, and initial classification steps were performed with two-fold binned data. False positives were removed in the first round of 2D classification. Remaining particles were subjected to several rounds of 2D classification, resulting in 680,465 particles that were further sorted in several rounds of 3D classification. The mTMEM16A cryo-EM map, low-pass filtered to 50 Å, was used as a reference for the first round of 3D classification and the best output class was used in subsequent jobs in an iterative way. The best 3D classes, comprising 219,302 particles, were subjected to auto-refinement, yielding a map with a resolution of 3.8 Å. In the last refinement iteration, a mask excluding the micelle was used and the refinement was continued until convergence, which

improved the resolution up to 3.5 Å. Using a mask generated from the final PDB model, we obtained a map at 3.4 Å. Finally, the newly available algorithms for CTF refinement and Bayesian polishing implemented in Relion3.0 (*Zivanov et al., 2018*)., where applied to further improve the resolution to 3.18 Å with a map sharpened at –94 Å$^2$. During final 3D classification and auto-refinement jobs, a C2-symmetry was imposed. Local resolution was estimated by RELION. All resolutions were estimated using the 0.143 cut-off criterion (*Rosenthal and Henderson, 2003*) with gold-standard Fourier shell correlation (FSC) between two independently refined half maps (*Scheres and Chen, 2012*). During post-processing, the approach of high-resolution noise substitution was used to correct for convolution effects of real-space masking on the FSC curve (*Chen et al., 2013*). The directional resolution anisotropy of density maps was quantitatively evaluated using 3DFSC (*Biyani et al., 2017*).

For the other datasets, a similar workflow for image processing was applied. In the case of the detergent dataset collected in absence of Ca$^{2+}$, a total of 1,314,676 particles were extracted after autopicking from 4621 images. Several rounds of 2D and 3D classification resulted in a final number of 194,284 particles, which after refinement yielded a 4.1 Å map. Providing a mask in the last iteration of the refinement, the resolution was improved to 3.8 Å. After post-processing the resolution improved to 3.6 Å. For the dataset in nanodiscs in the presence of Ca$^{2+}$, 1,019,012 particles were picked from 4480 images and extracted with a box size of 256 pixels. The final set of 186,487 particles was refined, yielding a map with a resolution of 3.9 Å, which after post-processing was improved to 3.5 Å. Finally, the dataset in nanodisc in absence of Ca$^{2+}$ resulted in 1,593,115 auto-picked particles from 6465 images, which were reduced to 280,891 particles after several rounds of 2D and 3D classification. Refinement was performed providing a mask in the last iteration, reaching a resolution of 3.3 Å after post-processing.

The analysis of structural heterogeneity within each dataset was conducted by running an additional 3D classification step with finer angular sampling on the final particle set. While some structural flexibility was revealed for α3 and α4 in the dataset in presence of Ca$^{2+}$ (*Figure 2—figure supplements 1I* and *3I*), no alternative conformations could be identified in the dataset in absence of Ca$^{2+}$ (*Figure 2—figure supplements 2I* and *4I*), as reflected in the better resolved density of the helices (*Figure 2—figure supplement 5*).

Finally, as revealed by the angular distribution and the 3D FSC plots obtained for the data in nanodiscs, the samples suffered from favorite orientation of the particles in vitreous ice, thereby compromising to some extent the detailed features of the maps (*Figure 2—figure supplements 3–5*). The problem is less pronounced in the nanodisc data obtained in presence of Ca$^{2+}$ than in the dataset of the Ca$^{2+}$-free sample. Nevertheless, the data in nanodiscs are of sufficient quality to allow for a comparison of corresponding states of mTMEM16F in detergent and a lipid environment.

## Model building refinement and validation

The model of the Ca$^{2+}$-bound state of mTMEM16F in digitonin was built in COOT (*Emsley and Cowtan, 2004*), using the structure of mTMEM16A (PDBID 5YOB) as template. The density was of sufficiently high resolution to unambiguously place the model consisting of residues 43–81, 87–149, 202–223, 228–427, 445–488, 503–587, 590–640, 645–791, 795–875. A DPPC (1,2-didecanoyl-sn-glycero-3-phosphocholine-ligand ID P1O) molecule was fitted into unassigned electron density close to the dimer interface of mTMEM16F, although the exact lipid species could not be determined. Coordinates were manually edited in COOT (*Emsley and Cowtan, 2004*) and the model was improved by real-space refinement in Phenix (*Adams et al., 2010*), whereby secondary structure elements and the symmetry between both subunits of the dimeric protein were constrained. The models of the Ca$^{2+}$-bound (PDBID 6QP6) and Ca$^{2+}$-free (PDBID 6QPB) structure in digitonin and the Ca$^{2+}$-bound structure obtained in nanodiscs (PDBID 6QPC) of mTMEM16F were built using the refined Ca$^{2+}$-bound structure as starting model. Major conformational changes were restricted to the second half of α6 and to some extent to α3 and α4. These regions were rebuilt manually in COOT (*Emsley and Cowtan, 2004*) and subsequently refined in Phenix (*Adams et al., 2010*), as described above. Due to the strong anisotropy of the dataset of mTMEM16F obtained in nanodiscs in absence of calcium, only the transmembrane domain, as a poly-alanine model, was interpreted (PDBID 6QPI). For validation of the refinement, Fourier shell correlations (FSC) between the refined model and the final map were determined (FSC$_{SUM}$, *Figure 2—figure supplements 1E*, *2E*, *3E* and *4E*). To monitor the effects of potential over-fitting, random shifts (up to 0.3 Å) were introduced into the coordinates of

the final model, followed by refinement in Phenix (*Adams et al., 2010*) against the first unfiltered half-map. The FSC between this shaken-refined model and the first half-map used during validation refinement is termed $FSC_{work}$, the FSC against the second half-map, which was not used at any point during refinement, $FSC_{free}$. The marginal gap between the curves describing $FSC_{work}$ and $FSC_{free}$ indicates no over-fitting of the model. The higher resolution of α-helix three obtained in the cryo-EM map of mTMEM16F in digitonin in absence of $Ca^{2+}$ allowed the unambiguous assignment of its residues. Based on this structure and the conservation between both paralogs, we remodeled α3 of mTMEM16A (*Paulino et al., 2017a*), which in its original data was poorly resolved. In the new model of mTMEM16A, the register of α3 has shifted by three residues and Arg 515 now superimposes with Arg 478 in mTMEM16F, both pointing into the pore lumen (*Figure 4B*; *Figure 4—figure supplement 1A*). The corrected model of mTMEM16A was used for structural comparisons throughout the manuscript. The pore diameter was calculated in HOLE (*Smart et al., 1996*). Pictures were generated using pymol (The PyMOL Molecular Graphics System, Version 2.0 Schrödinger, LLC), chimera (*Pettersen et al., 2004*) and chimeraX (*Goddard et al., 2018*).

## Electrostatic calculations

The electrostatic potential was calculated as described (*Paulino et al., 2017a*). The linearized Poisson–Boltzmann equation was solved in CHARMM (*Brooks et al., 1983*; *Im et al., 1998*) on a 200 Å ×140 Å × 190 Å grid (1 Å grid spacing) followed by focusing on a 135 Å x 100 Å x 125 Å grid (0.5 Å grid spacing). Partial protein charges were derived from the CHARMM36 all-hydrogen atom force field. Hydrogen positions were generated in CHARMM. The protein was assigned a dielectric constant ($\epsilon$) of 2. Its transmembrane region was embedded in a 30-Å-thick slab ($\epsilon$ = 2) representing the hydrophobic core of the membrane and two adjacent 15-Å-thick regions ($\epsilon$ = 30) representing the headgroups. This region contained a cylindrical hole around the water-filled intracellular vestibule of one subunit and was surrounded by an aqueous environment ($\epsilon$ = 80) containing 150 mM of monovalent mobile ions.

## Statistics and reproducibility

Electrophysiology data were repeated multiple times from different transfections with very similar results. Conclusions of experiments were not changed upon inclusion of further data. In all cases, leaky patches were discarded. Statistical difference between the constructs were determined with one-way ANOVA and Tukey-Kramer post-hoc test.

# Acknowledgments

We thank S Klauser, S Rast and M Punter for their help in establishing and maintaining the computer infrastructure. Y Neldner for help during the generation of stable mTMEM16F cell-lines. D Deneka is acknowledged for help with nanodisc reconstitution. V Kalienkova is acknowledged for helpful discussions. All members of the Dutzler and Paulino labs are acknowledged for their help at various stages of the project.

# Additional information

## Funding

| Funder | Grant reference number | Author |
| --- | --- | --- |
| Nederlandse Organisatie voor Wetenschappelijk Onderzoek | 740.018.016 | Cristina Paulino |
| FP7 European Research Council | 339116, AnoBest | Raimund Dutzler |

The funders had no role in study design, data collection and interpretation, or the decision to submit the work for publication.

## Author contributions

Carolina Alvadia, Formal analysis, Validation, Investigation, Visualization, Writing—original draft, Writing—review and editing, Generated stable cell-lines and expression constructs, Purified proteins for cryo-EM and functional characterization, Reconstituted protein into nanodiscs and liposomes and carried out lipid transport experiments, Helped in model building and refinement, Jointly planned experiments, analyzed the data and wrote the manuscript; Novandy K Lim, Formal analysis, Validation, Investigation, Writing—original draft, Writing—review and editing, Purified proteins for cryo-EM and functional characterization, Recorded and analyzed electrophysiology data, Jointly planned experiments, analyzed the data and wrote the manuscript; Vanessa Clerico Mosina, Formal analysis, Validation, Investigation, Visualization, Writing—original draft, Writing—review and editing, Collected cryo-EM data, Carried out image processing, Jointly planned experiments, analyzed the data and wrote the manuscript; Gert T Oostergetel, Formal analysis, Collected cryo-EM data; Raimund Dutzler, Conceptualization, Resources, Formal analysis, Supervision, Funding acquisition, Validation, Visualization, Writing—original draft, Project administration, Writing—review and editing, Jointly planned experiments, analyzed the data and wrote the manuscript; Cristina Paulino, Conceptualization, Resources, Formal analysis, Supervision, Funding acquisition, Validation, Investigation, Visualization, Writing—original draft, Project administration, Writing—review and editing, Prepared the samples for cryo-EM, Collected cryo-EM data, Carried out image processing, Performed model building and refinement, Jointly planned experiments, analyzed the data and wrote the manuscript

## Author ORCIDs

Carolina Alvadia (iD) http://orcid.org/0000-0001-8446-1098
Novandy K Lim (iD) http://orcid.org/0000-0001-5098-929X
Vanessa Clerico Mosina (iD) https://orcid.org/0000-0001-8013-0144
Gert T Oostergetel (iD) http://orcid.org/0000-0001-6816-136X
Raimund Dutzler (iD) http://orcid.org/0000-0002-2193-6129
Cristina Paulino (iD) http://orcid.org/0000-0001-7017-109X

## Decision letter and Author response

Decision letter https://doi.org/10.7554/eLife.44365.045
Author response https://doi.org/10.7554/eLife.44365.046

# Additional files

## Supplementary files

• Transparent reporting form
DOI: https://doi.org/10.7554/eLife.44365.027

## Data availability

The three-dimensional cryo-EM density maps of calcium-bound mTMEM16F in detergent and nanodiscs have been deposited in the Electron Microscopy Data Bank under accession numbers EMD-4611 and EMD-4613, respectively. The maps of calcium-free samples in detergent and nanodiscs were deposited under accession numbers EMD-4612 and EMD-4614, respectively. The deposition includes the cryo-EM maps, both half-maps, and the mask used for final FSC calculation. Coordinates of all models have been deposited in the Protein Data Bank under accession numbers 6QP6 ($Ca^{2+}$-bound, detergent), 6QPC ($Ca^{2+}$-bound, nanodisc), 6QPB ($Ca^{2+}$-free, detergent) and 6QPI ($Ca^{2+}$-free, nanodisc).

The following datasets were generated:

| Author(s) | Year | Dataset title | Dataset URL | Database and Identifier |
|---|---|---|---|---|
| Alvadia C, Lim NK, Clerico Mosina V, Oostergetel GT, Dutzler R, Paulino C | 2019 | Cryo-EM structure of calcium-bound mTMEM16F lipid scramblase in digitonin | https://www.rcsb.org/structure/6QP6 | Protein Databank, 6QP6 |
| Alvadia C, Lim NK, | 2019 | Cryo-EM structure of calcium-free | https://www.rcsb.org/ | Protein Databank, |

| Clerico Mosina V, Oostergetel GT, Dutzler R, Paulino C | | mTMEM16F lipid scramblase in digitonin | structure/6QPB | 6QPB |
|---|---|---|---|---|
| Alvadia C, Lim NK, Clerico Mosina V, Oostergetel GT, Dutzler R, Paulino C | 2019 | Cryo-EM structure of calcium-bound mTMEM16F lipid scramblase in nanodisc | https://www.rcsb.org/structure/6QPC | Protein Databank, 6QPC |
| Alvadia C, Lim NK | 2019 | Cryo-EM structure of calcium-free mTMEM16F lipid scramblase in nanodisc | https://www.rcsb.org/structure/6QPI | Protein Databank, 6QPI |
| Alvadia C, Lim NK, Clerico Mosina V, Oostergetel GT, Dutzler R, Paulino C | 2019 | Cryo-EM structure of calcium-bound mTMEM16F lipid scramblase in digitonin | http://www.ebi.ac.uk/pdbe/entry/emdb/EMD-4611 | Electron Microscopy Data Bank, EMD-4611 |
| Alvadia C, Lim NK, Clerico Mosina V, Oostergetel GT, Dutzler R | 2019 | Cryo-EM structure of calcium-free mTMEM16F lipid scramblase in digitonin | http://www.ebi.ac.uk/pdbe/entry/emdb/EMD-4612 | Electron Microscopy Data Bank, EMD-4612 |
| Alvadia C, Lim NK, Clerico Mosina V, Oostergetel GT, Dutzler R, Paulino C | 2019 | Cryo-EM structure of calcium-bound mTMEM16F lipid scramblase in nanodisc | http://www.ebi.ac.uk/pdbe/entry/emdb/EMD-4613 | Electron Microscopy Data Bank, EMD-4613 |
| Alvadia C, Lim NK, Clerico Mosina V, Oostergetel GT, Dutzler R, Paulino C | 2019 | Cryo-EM structure of calcium-free mTMEM16F lipid scramblase in nanodisc | http://www.ebi.ac.uk/pdbe/entry/emdb/EMD-4614 | Electron Microscopy Data Bank, EMD-4614 |

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
