## [Decision Letter]

Thank you for submitting your article "Cryo-EM structures and functional characterization of the lipid scramblase TMEM16F" for consideration by *eLife*. Your article has been reviewed by Richard Aldrich as the Senior Editor, Kenton Swartz as the Reviewing Editor, and three reviewers. The following individuals involved in review of your submission have agreed to reveal their identity: Joel Meyerson (Reviewer #1); Angela Ballesteros (Reviewer #2); Stephen Barstow Long (Reviewer #3).

The reviewers have discussed the reviews with one another and the Reviewing Editor has drafted this decision to help you prepare a revised submission.

Summary:

The TMEM16 family of proteins contains members who function either as lipid scramblases, or anion selective channels. TMEM16F is a lipid scramblase integral to the coagulation response in platelets, and splicing mutations that yield a truncated protein cause a bleeding disorder in humans. Additionally, TMEM16F shows rapidly activated calcium-dependent currents. The authors' central aim is to understand the mechanism(s) underpinning the protein's dual lipid and ion conduction abilities.

To pursue this question Alvadia and coworkers determined cryo-EM structures of mTMEM16F with and without calcium, solubilized in both digitonin and nanodisc. These four structures were used for analysis throughout. For comparative structural analysis, mTMEM16F structures were compared to existing structures of nhTMEM16 (scramblase) and mTMEM16A (anion channel). The mTMEM16F calcium-free structures reveal a closed conformation as expected, but the calcium-bound structures do not reveal an active state conformation. However, the calcium-bound structure in nanodisc, in particular, shows conformational mobility for α-helices 3 and 4. The authors infer that the calcium-bound structure perhaps represents an intermediate state.

In parallel, the authors reconstituted mTMEM16F in liposomes to assay its lipid transport properties and used patch clamp recordings to characterize the protein's ion conduction properties. On this functional platform they were able to interrogate mutations targeted at calcium binding (E667Q, E624Q), ion conduction (Y563S, Q559K, R478A), and scrambling (K370A, Q623F, TMEM16F^scrd^ chimera). By integrating the structural and functional analysis, the authors ultimately propose a model in which both transport processes are activated by the same mechanism and mediated by distinct protein conformations that exist in equilibrium. This study succeeds in defining the architecture of mTMEM16F and performing functional experiments to probe the structures. The cryo-EM work is expertly done, the text is well-written, and figures are clear and professional. While there remains uncertainty about the correspondence of the calcium-bound conformation of the protein to a defined functional state, the authors openly acknowledge this, address it to the best of their ability within the scope of the paper, and fairly present it as a question that begs further interrogation. The current study tackles an experimentally challenging member of an important family of proteins, enhances our understanding of lipid scramblase and ion channel activity of TMEM16F, and provides the foundation for future studies. The following are suggestions for improving the manuscript in revision.

Essential revisions:

1) For experiments examining calcium-activated currents, in addition to providing representative current traces at different calcium concentrations, please indicate the zero current level and provide data for 0 calcium so the reader can appreciate whether there is constitutive activation. It would also be good to provide current traces for Y363S and Q559K.

2) The calculation of the rectification index for the wild-type TMEM16F and Y563S, Q559K and R478A mutants shown in Figure 6C and Figure 6—figure supplement 2A is hard to comprehend and important methodological details about these experiments are missing and make the evaluation and interpretation of these data difficult.

- Please provide more detailed information about the protocol used in the recordings and the calculation of this index should be included in the Materials and methods section.

- Including the full traces at +80 and -80mV in the figures and indicating which point of the trace was used to estimate both the instantaneous and steady-state rectification indexes would help the reader to better understand the data.

- The differences observed in the rectification index of the steady-state currents with the mutants Q559K and R478A are not clearly explained in the text (subsection “Functional characterization of mutants affecting ion conduction and scrambling”), and the statistical evaluation on the significance of these differences is not presented in the figure. Please revise the text and include the statistical significances in the figure and/or figure legend.

3) The ion selectivity properties are only estimated for the mutant Q559K (Figure 6—figure supplement 2 panel B). Have the authors also evaluated the ionic selectivity of the Y563S mutant, which is also located in the pore constriction?

4) The authors show in this study that the calcium-activated currents on mutants for the residues located at the pore constriction (Y563S) and the intracellular vestibule close to the neck (Q559K) are activated at lower calcium concentration that the wild-type TMEM16F and could play a role in the stabilization of the open state. Have the authors undertaken scrambling assays with the TMEM16F Y563S and Q559K mutants to evaluate the role of the pore region in lipid scrambling? This might help to distinguish between some of the alternative mechanisms discussed in the manuscript.

5) In TMEM16A, the glycine residue at the hinge of the TM6 (G644, Paulino et al., 2017 and Lam et al., 2018) allows the movement of TM6 after calcium release and the partial closure of the cavity at the intracellular side. The authors show that the G615A mutant causes a left shift in the EC50 for calcium activation, suggesting a displacement of the open-close equilibrium towards the open state. The evaluation of the lipid scrambling of the TMEM16F G615A mutant would highlight the role of this residue in PLS activity and help to investigate whether the activation of the scramblase and channel functions share a common mechanism (subsection “Functional characterization of mutants of the Ca^2+^-binding site”). Do the authors have any data on this?

---

## [Author Response]

Essential revisions:1) For experiments examining calcium-activated currents, in addition to providing representative current traces at different calcium concentrations, please indicate the zero current level and provide data for 0 calcium so the reader can appreciate whether there is constitutive activation. It would also be good to provide current traces for Y363S and Q559K.

We now provide the current traces of the mutants Y363S and Q559K, indicate the 0-current levels in all traces and provide an inset of each figure that enlarges the 0 Ca^2+^ currents that have already been included in the figure. In no case did we observe significant activity in absence of Ca^2+^. See Figure 6—figure supplement 2.

2) The calculation of the rectification index for the wild-type TMEM16F and Y563S, Q559K and R478A mutants shown in Figure 6C and Figure 6—figure supplement 2A is hard to comprehend and important methodological details about these experiments are missing and make the evaluation and interpretation of these data difficult.- Please provide more detailed information about the protocol used in the recordings and the calculation of this index should be included in the Materials and methods section.

We have included a short paragraph in the subsection “Electrophysiology” describing the quantification of the rectification index. “The rectification indices of instantaneous currents were calculated as the ratio between the current measured at 80 mV, 100 ms before the voltage jump, and 6 ms after changing to -80 mV at which the capacitive transients have already decayed. For calculation of the rectification index of steady-state currents, the value at -80 mV was taken once the current response has reached a plateau (typically 150-200 ms after the voltage change).”

- Including the full traces at +80 and -80mV in the figures and indicating which point of the trace was used to estimate both the instantaneous and steady-state rectification indexes would help the reader to better understand the data.

We have added this to Figure 6—figure supplement 2.

- The differences observed in the rectification index of the steady-state currents with the mutants Q559K and R478A are not clearly explained in the text (subsection “Functional characterization of mutants affecting ion conduction and scrambling”), and the statistical evaluation on the significance of these differences is not presented in the figure. Please revise the text and include the statistical significances in the figure and/or figure legend.

The change in the steady state rectification was generally more severe for the mutant R478A compared to the mutant Q559K. We have rewritten the paragraph as follow (see subsection “Functional characterization of mutants affecting ion conduction and scrambling”):”In contrast to the subtle phenotype of Y563S, the mutation of the conserved Arg 478 (Figure 4B) at the extracellular entrance of the narrow neck to alanine, yields a several-fold decreased current density compared to WT, yet with increased potency of Ca^2+^ and strongly decreased voltage-dependent relaxations. Consequently, the strong rectification at steady-state disappears, suggesting that the residue plays an important role during channel activation (Figure 6E; Figure 6—figure supplement 2A,F). Collectively, the observed changes in the activation and conduction properties of the mutants Q559K and R478A are consistent with their location in the pore. The increase of Ca^2+^-potency found in R478A and the strongly decreased time-dependent reduction of currents at negative voltage might indicate a stabilization of the open state. Moreover, the change in selectivity observed for the mutant Q559K provides direct evidence for its location at the pore lining.”

We have also carried out a statistical analysis to show that the change in R478A significant but not the change in Q559K due to the high variance of the data. We now state this in the methods and the figure legend. See subsection “Electrophysiology”: “For the characterization of rectification properties of steady-state currents, only the shift of the mutant R478A compared to WT was calculated to be statistically significant. In case of the mutant Q559K, the high variance due to the presence of two populations of data with strong and weak rectification properties rendered the shift in its value as statistically insignificant.”

See also Figure 6 legend: “Unlike Q599K, the difference between R478A and WT was found to be statistically significant.”

3) The ion selectivity properties are only estimated for the mutant Q559K (Figure 6—figure supplement 2 panel B). Have the authors also evaluated the ionic selectivity of the Y563S mutant, which is also located in the pore constriction?

We have now also estimated the ion selectivity of Y563S and did not find any significant change compared to WT. This is expected for a channel that does not strongly interact with permeating ions.

4) The authors show in this study that the calcium-activated currents on mutants for the residues located at the pore constriction (Y563S) and the intracellular vestibule close to the neck (Q559K) are activated at lower calcium concentration that the wild-type TMEM16F and could play a role in the stabilization of the open state. Have the authors undertaken scrambling assays with the TMEM16F Y563S and Q559K mutants to evaluate the role of the pore region in lipid scrambling? This might help to distinguish between some of the alternative mechanisms discussed in the manuscript.

We have remeasured the concentration-response relationships of currents of the mutants Y563S and Q559K and found no significant shift to WT (although we could reproduce the lower slope in the activation of Q559K) and thus have removed the claim that both mutants stabilize the open conformation. In contrast we still find the increase in the Ca^2+^-potency of R478A and TMEM16F^SCRD^. In our revised manuscript we have now included data on the scrambling properties of Y563S and Q559K and found that both mutants show robust scrambling activity, in contrast to R478A and TMEM16F^SCRD^. These results now even put stronger emphasis on the reversed phenotype of R478A and TMEM16F^SCRD^ with respect to ion conduction and scrambling.

We have introduced changes in several parts of the manuscript: see subsection “Functional characterization of mutants affecting ion conduction and scrambling” and the Discussion section.

5) In TMEM16A, the glycine residue at the hinge of the TM6 (G644, Paulino et al., 2017 and Lam et al., 2018) allows the movement of TM6 after calcium release and the partial closure of the cavity at the intracellular side. The authors show that the G615A mutant causes a left shift in the EC50 for calcium activation, suggesting a displacement of the open-close equilibrium towards the open state. The evaluation of the lipid scrambling of the TMEM16F G615A mutant would highlight the role of this residue in PLS activity and help to investigate whether the activation of the scramblase and channel functions share a common mechanism (subsection “Functional characterization of mutants of the Ca^2+^-binding site”). Do the authors have any data on this?

We should emphasize that, due to technical limitations of the assay, the quantification of dose-response relationships from scrambling data is much less accurate than for ion-conduction. This does not affect the mutant E667Q where the Ca^2+^-potency is 100-fold lower than for WT and where scrambling data and ion conduction data show comparable shifts. In contrast, the shifts for the mutant G615A observed for ionic currents is below 10-fold, which can still be accurately determined by electrophysiology but not from scrambling data. Preliminary scrambling data for G615A shows a Ca^2+^-dependent activation that is not significantly different from WT. Due the described limitations of the assay we prefer not to include these measurements in our study since it would lead to misleading interpretations.